# HARNESSING EXPLANATIONS: LLM-TO-LM INTERPRETER FOR ENHANCED TEXT-ATTRIBUTED GRAPH REPRESENTATION LEARNING

**Xiaoxin He[1], Xavier Bresson[1], Thomas Laurent[2], Adam Perold[3], Yann LeCun[4,5], Bryan Hooi[1]**
[1]National University of Singapore, [2]Loyola Marymount University
[3]Element, Inc., [4]New York University, [5]Meta AI
`{xiaoxin, xaviercs, bhooi}@comp.nus.edu.sg, tlaurent@lmu.edu`
`ap@elementresearch.com, yann@cs.nyu.edu`

## ABSTRACT

Representation learning on text-attributed graphs (TAGs) has become a critical research problem in recent years. A typical example of a TAG is a paper citation graph, where the text of each paper serves as node attributes. Initial graph neural network (GNN) pipelines handled these text attributes by transforming them into shallow or hand-crafted features, such as skip-gram or bag-of-words features. Recent efforts have focused on enhancing these pipelines with language models (LMs), which typically demand intricate designs and substantial computational resources. With the advent of powerful large language models (LLMs) such as GPT or Llama2, which demonstrate an ability to reason and to utilize general knowledge, there is a growing need for techniques which combine the textual modelling abilities of LLMs with the structural learning capabilities of GNNs. Hence, in this work, we focus on leveraging LLMs to capture textual information as features, which can be used to boost GNN performance on downstream tasks. A key innovation is our use of *explanations as features*: we prompt an LLM to perform zero-shot classification, request textual explanations for its decision-making process, and design an *LLM-to-LM interpreter* to translate these explanations into informative features for downstream GNNs. Our experiments demonstrate that our method achieves state-of-the-art results on well-established TAG datasets, including `Cora`, `PubMed`, `ogbn-arxiv`, as well as our newly introduced dataset, `tape-arxiv23`. Furthermore, our method significantly speeds up training, achieving a 2.88 times improvement over the closest baseline on `ogbn-arxiv`. Lastly, we believe the versatility of the proposed method extends beyond TAGs and holds the potential to enhance other tasks involving graph-text data [1].

## 1 INTRODUCTION

Many real-world graphs possess textual information, and are often referred to text-attributed graphs (Yang et al., 2021). In TAGs, nodes typically represent text entities, such as documents or sentences, while edges signify relationships between these entities. For example, the `ogbn-arxiv` dataset (Hu et al., 2020a) represents a citation network in TAG form, where each node corresponds to a paper, with its title and abstract serving as node attributes. More generally, the combination of textual attributes with graph topology provides a rich source of information, significantly enhancing representation learning for important applications, such as text classification (Yang et al., 2015; Wang et al., 2016; Yasunaga et al., 2017; Chien et al., 2021; Zhao et al., 2022), recommendation systems (Zhu et al., 2021), social networks, and fake news detection (Liu et al., 2019).

**Representation learning on TAGs.** Prior research has explored various approaches for representation learning on TAGs. The standard GNN pipeline (illustrated in Figure 1 in light yellow), first encodes the textual attributes of each node using shallow or hand-crafted features such as skip-gram (Mikolov et al., 2013) or bag-of-words (BoW) (Harris, 1985) (refer to Table 9). The result-

---

[1]Our codes and datasets are available at: `https://github.com/XiaoxinHe/TAPE`

**Overview of TAPE:** **T**itle, **A**bstract, **P**rediction & **E**xplanation

*Step 1: Node Feature Extraction*

*Step 2: Downstream Tasks on TAGs*

**Prediction**: cs.CV, cs.IR, cs.CL, cs.LG, cs.AI.

**Abstract**: Text in curve orientation, despite being one of the common text orientations in real world environment...
**Title**: Total Text A Comprehensive Dataset For Scene Text Detection And Recognition.
**Question:** Which arXiv CS sub-category does this paper belong to? Give 5 likely arXiv CS sub-categories as a comma-separated list ordered from most to least likely, in the form "cs.XX", and provide your reasoning.
**Answer**:

**Explanation**: The paper is about a new dataset for scene text detection and recognition, which is a topic related to computer vision (cs.CV). The paper also mentions the use of deep learning techniques such as DeconvNet, which falls under the sub-category of artificial intelligence (cs.AI). The dataset is annotated and involves text recognition, which could also fall under the sub-categories of information retrieval (cs.IR) and natural language processing (cs.CL). Finally, the paper discusses the effectiveness of different solutions, which could be evaluated using machine learning techniques, falling under the sub-category of machine learning (cs.LG).

Figure 1: Our framework leverages large language models (LLMs) to enhance representation learning on TAGs. First, textual attributes of each node, *i.e.,* title and abstract, are wrapped in a custom prompt (green box) to query the LLM, here GPT-3.5 (Brown et al., 2020), which generates a ranked prediction list and explanation (yellow box). Next, the original text, predictions, and explanation are used to fine-tune a language model (LM), here DeBERTa (He et al., 2021), and then transformed into vectorial node features. Finally, these enriched node features, *i.e.,* $h_{\text{orig}}$, $h_{\text{expl}}$ and $h_{\text{pred}}$, are used in any downstream GNN, *e.g.,* RevGAT (Li et al., 2021) to predict unknown node classes.

ing node features are then used as input for a GNN. For instance, the Open Graph Benchmark (OGB) (Hu et al., 2020a) generated BoW and skip-gram (Mikolov et al., 2013) features for the `ogbn-products` and `ogbn-arxiv` datasets respectively. These processed features are readily available within popular graph libraries, such as PyTorch Geometric (PyG) (Fey & Lenssen, 2019) and Deep Graph Library (DGL) (Wang et al., 2019), and have been widely used by the graph community. However, these shallow text embeddings are limited in the complexity of the semantic features they can capture, especially when compared to approaches based on multi-layer LMs.

**LM-based pipeline for TAGs.** Recent works have therefore focused on designing LM-based techniques to better capture the context and nuances of text within TAGs (Chien et al., 2021; Zhao et al., 2022; Dinh et al., 2022). In this approach, pre-trained LMs are fine-tuned and used to generate node embeddings that are tailored to the specific TAG tasks (depicted in Figure 1 in light gray). For example, Chien et al. (2021) fine-tuned an LM using a neighborhood prediction task, while Zhao et al. (2022) fine-tuned an LM to predict the label distribution from a GNN's outputs. LM-based models have achieved state-of-the-art (SOTA) results in node classification on `ogbn-arxiv` and `ogbn-products` (Zhao et al., 2022). However, these works typically entail intricate designs and demand substantial computational resources. Furthermore, for scalability reasons, existing works mostly rely on relatively small LMs, such as BERT (Devlin et al., 2018) and DeBERTa (He et al., 2021), and thus lack the complex reasoning abilities associated with larger language models.

**Large Language Models.** The advent of large pre-trained models, exemplified by GPT (Brown et al., 2020), has revolutionized the field of language modeling. LLMs have notably enhanced performance across various natural language processing (NLP) tasks, and enabled sophisticated language processing capabilities such as complex and zero-shot reasoning. Furthermore, scaling laws (Kaplan et al., 2020) have revealed predictable rules for performance improvements with model and training data size. Additionally, LLMs have exhibited "emergent abilities" that were not explicitly trained for, such as arithmetic, multi-step reasoning and instruction following (Wei et al., 2022). While LLMs have found new success in domains like computer vision (Tsimpoukelli et al., 2021), their potential benefits when applied to TAG tasks remain largely uncharted. This presents an exciting and promising avenue for future research, and it is precisely this untapped potential that we aim to explore in this work.

**LMs vs. LLMs.** In this paper, we make a clear distinction between "LMs" and "LLMs". We use LMs to refer to relatively small language models that can be trained and fine-tuned within the constraints of *an academic lab budget*. We refer to LLMs as very large language models that are capable of learning significantly more complex linguistic patterns than LMs, such as GPT-3/4. These models typically have tens or hundreds of billions of parameters and require *substantial computational resources* to train and use, *e.g.,* GPT-3 was trained on a supercomputer with 10,000 GPUs. The size and complexity of recent LLMs have raised concerns about their scalability, as they can be too large even to run inference on the machines typically available within academic research labs. To address this issue, LLMs are often made accessible through *language modeling as a service* (LMaaS) (Sun et al., 2022). This approach enables developers to harness the power of LLMs without necessitating extensive computational resources or specialized expertise. In the context of this paper, one of our primary objectives is to extract information from an LLM in a LMaaS-compatible manner. As a result, we do not require fine-tuning the LLM or extracting its logits; rather, we focus solely on obtaining its output in textual form. In contrast, existing LM-based techniques (Chien et al., 2021; Zhao et al., 2022; Dinh et al., 2022) are not directly compatible with LLMs, as they require fine-tuning of LMs, as well as accessing their latent embeddings or logits, which GPT-3/4 do not provide. Consequently, to the best of our knowledge, the use of LLMs in TAG tasks remains unexplored.

**Preliminary study.** To assess the potential of LLMs in enhancing representation learning for TAGs, we conducted an initial investigation into leveraging GPT-3.5 for zero-shot classification on the `ogbn-arxiv` dataset. Using task-specific prompts consisting of paper titles, abstracts, and questions, GPT-3.5 achieved a promising accuracy of 73.5%, along with high-quality text explanations, surpassing several fully trained GNN baselines like RevGAT (Li et al., 2021) with OGB features (70.8% accuracy), but falling short of the SOTA accuracy of 76.6% (Zhao et al., 2022).

**The present work: LLM augmentation using explanations.** We introduce a novel framework that leverages LLMs to improve representation learning on TAGs. A key innovation is the concept of *explanations as features*. By prompting a powerful LLM to explain its predictions, we extract its relevant prior knowledge and reasoning steps, making this information digestible for smaller models, akin to how human experts use explanations to convey insights. To illustrate this concept further, observe in Figure 1 that the explanations (in the yellow box) highlight and expand upon key crucial information from the text, such as "deep learning techniques such as DeconvNet," and the relationship between text recognition and information retrieval. These explanations draw from the LLM's general knowledge and serve as valuable features for enhancing subsequent TAG pipeline phases. In practice, we design a tailored prompt to query an LLM such as GPT or Llama2 to generate both a *ranked prediction list* and a *textual explanation* for its predictions. These predictions and explanations are then transformed into informative node features through fine-tuning a smaller LM such as DeBERTa (He et al., 2021) for the target task, providing tailored features for any downstream GNNs. This smaller model acts as an *interpreter*, facilitating seamless communication between the LLM (handling text) and the GNN (managing vectorial representation).

Our contributions are summarized as follows:

- **Novel LMaaS-compatible approach.** We propose the first LMaaS-compatible approach, to the best of our knowledge, for leveraging LLMs to enhance representation learning on TAGs. Our innovations involve extracting explanations from an LLM, here GPT-3.5 and Llama2, and subsequently employing an LLM-to-LM interpreter to translate textual explanations into enriched node vector representations for downstream GNNs. Our approach improves modularity and efficiency compared to prior LM+GNN models.

- **SOTA performance.** Extensive experiments demonstrate that our method significantly boost the performance of various GNN models across diverse datasets. Notably, we achieve top-1 performance on `ogbn-arxiv` with significantly lower computation time, *i.e.,* $2.88\times$ faster than GLEM, and also excel in the TAG versions of `PubMed` and `Cora` datasets.

- **Data contribution.** We provide open-source access to our codes, pre-trained networks and enriched features. Additionally, recognizing the absence of raw text data for `Cora` and `PubMed` in common repositories (*e.g.,* PyG, DGL), we have collected and released these datasets in TAG format. Furthermore, we introduce the new `tape-arxiv23` citation graph dataset, extending beyond GPT-3's knowledge cutoff, *i.e.,* Sept. 2021. These datasets can serve as valuable resources for the NLP and GNN research community.

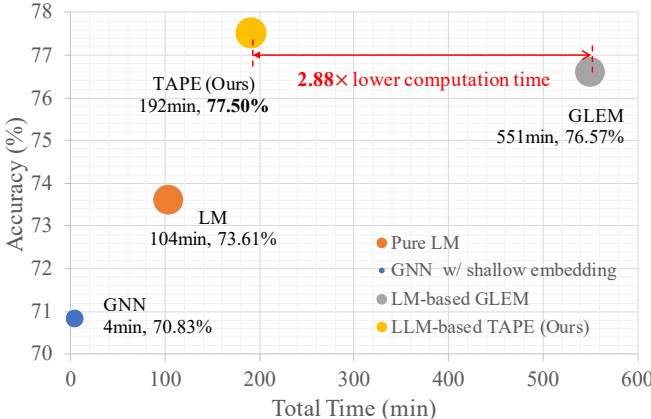

Figure 2: The performance trade-off between node classification accuracy and total training time on `ogbn-arxiv` (Hu et al., 2020a) for various training approaches that combine language models (LMs) and graph neural networks (GNNs). The experiment employs DeBERTa-base (He et al., 2021) as the LM backbone and RevGAT (Li et al., 2021) as the GNN backbone, with the size of the marker indicating the number of parameters.

## 2 RELATED WORK

**Shallow embedding pipeline for TAGs.** In the context of learning representations on TAGs, a common approach involves combining graph-based learning with language modeling techniques. One prevalent strategy is to transform text attributes into shallow or hand-crafted features, such as skip-gram (Mikolov et al., 2013) or BoW (Harris, 1985) features. Detailed information is available in Table 9. These engineered features can then be fed as inputs to a graph-based learning algorithm, such as a graph convolutional network (GCN) (Kipf & Welling, 2016), which learns embeddings capturing the graph structure while incorporating the extracted text features. Shallow embedding methods are widely used in the graph community due to their simplicity and computational efficiency, such as for designing GNN architectures (Veličković et al., 2017; Chiang et al., 2019; Velickovic et al., 2019; Zhang et al., 2021) or benchmarking graph learning (Yang et al., 2016; Hu et al., 2020a). However, they may have limitations in capturing complex semantic relationships and fully leveraging the richness of text attributes, particularly in scenarios involving intricate semantic relationships and contextual information.

**LM-based pipeline for TAGs.** To overcome the limitations of shallow embedding approaches, researchers have explored deep embedding techniques by fine-tuning pre-trained LMs, such as BERT (Devlin et al., 2018), to generate node embeddings that are specifically adapted to the domain and context of the TAGs. These deep embeddings effectively capture the semantic richness of text attributes, leading to improved performance on various TAG-related tasks. Integrating LM-based embeddings and graph-based learning can be done through different approaches. One approach is to use a cascaded architecture, where the node features are first encoded independently by the LMs, and then fed into GNN models. This representation paradigm has been widely adopted in subsequent works, such as TextGNN (Zhu et al., 2021), GIANT (Chien et al., 2021), GPT-GNN (Hu et al., 2020b), SimTeg (Duan et al., 2023), as well as in studies related to knowledge graphs (Yasunaga et al., 2021; Zhang et al., 2022) and fact verification (Liu et al., 2019; Zhou et al., 2019) that are beyond the scope of this work. An alternative approach involves fusing text encoding and graph aggregation into an iterative workflow, enabling the model to refine both the text representations and the node embeddings simultaneously, such as Graphormer (Yang et al., 2021), DRAGON (Yasunaga et al., 2022), and GLEM (Zhao et al., 2022), to name a few.

**LLM-based pipeline for TAGs.** Incorporating LLMs into TAG tasks presents a promising frontier. LLMs such as ChatGPT (Brown et al., 2020) by OpenAI, PaLM (Chowdhery et al., 2022) by Google, and LLaMA (Touvron et al., 2023) by Meta, have demonstrated their effectiveness across a spectrum of NLP tasks. However, their potential benefits for TAG tasks have yet to be fully explored. While some recent research efforts have sought to evaluate the capacity of LLMs in understanding graph-structured data and enhance their graph processing capabilities (Wang et al., 2023; Zhang, 2023; Guo et al., 2023), these endeavors, while valuable, may not be directly aligned with our specific focus

on TAGs. By exploring LLM-based methods designed specifically for TAGs, we can unlock new possibilities for improving TAG prediction performance and advancing our understanding of text attributes within graph-based data. Notably, our initial attempt has already inspired further research endeavors in this direction.

## 3 FORMALIZATION

In this section, we introduce notation and formalize some concepts related to language models, large language models, and graph neural networks for node classification on TAGs.

**Text-attributed graphs.** Formally, a TAG can be represented as $\mathcal{G} = (\mathcal{V}, A, \{s_n\}_{n \in \mathcal{V}})$, where $\mathcal{V}$ is a set of $N$ nodes, $A \in \mathbb{R}^{N \times N}$ is the adjacency matrix, and $s_n \in \mathcal{D}^{L_n}$ is a sequential text associated with node $n \in \mathcal{V}$, with $\mathcal{D}$ as the words or tokens dictionary, and $L_n$ as the sequence length. In this paper, we investigate node classification on TAGs. Specifically, given some labeled nodes $\mathcal{L} \subset \mathcal{V}$, the goal is to predict the labels of the remaining unlabeled nodes $\mathcal{U} = \mathcal{V} \setminus \mathcal{L}$.

**Language models for text classification.** In the context of TAGs, LMs can be employed to encode the text attributes associated with each node and learn a representation that captures the semantic meaning of the text. Let $s_n \in \mathcal{D}^{L_n}$ denote the text attributes of node $n$, and LM be a pre-trained network, such as BERT (Devlin et al., 2018) or DeBERTa (He et al., 2021). Then, the text attributes of node $n$ can be encoded by applying the LM to $s_n$ as follows:

$$h_n = \text{LM}(s_n) \in \mathbb{R}^d, \tag{1}$$

where $h_n$ is the output of the LM, and $d$ is the dimension of the output vector.

To perform node classification, the output is employed as input to a classifier, such as a logistic regression or a neural network. The goal is to learn a function that maps the encoded text attributes to the corresponding node labels.

**Large language models and prompting.** LLMs have introduced a new paradigm for task-adaptation known as "pre-train, prompt, and predict", replacing the traditional "pre-train, fine-tune" procedure. In this paradigm, the LLM is first pre-trained on a large corpus of text data to learn general language representations. Then, rather than fine-tuning the model on task-specific labeled data, the model is prompted with a natural language prompt that specifies the task and context, and the model generates the output directly based on the prompt and the input (Liu et al., 2023).

The prompt can take various forms, such as a single sentence or a longer passage, and can include additional information or constraints to guide the model's behavior. Let $\mathcal{M}$ be an LLM that takes as input a sequence of tokens $x = (x_1, x_2, \ldots, x_q)$ and produces as output a sequence of tokens $y = (y_1, y_2, \ldots, y_m)$. The model $\mathcal{M}$ is typically trained to optimize a conditional probability distribution $p(y|x)$, which assigns a probability to each possible output sequence $y$ given $x$. To include a prompt $p$ with the input sequence $x$, we can concatenate them into a new sequence $\hat{x} = (p, x_1, x_2, \ldots, x_q)$. We then use $\hat{x}$ to compute the conditional probability distribution $p(y|\hat{x})$. Formally, the probability of the output sequence $y$ given $\hat{x}$ is:

$$p(y|\hat{x}) = \prod_{i=1}^{m} p(y_i|y_{<i}, \hat{x}), \tag{2}$$

where $y_{<i}$ represents the prefix of sequence $y$ up to position $i-1$, and $p(y_i|y_{<i}, \hat{x})$ represents the probability of generating token $y_i$ given $y_{<i}$ and $\hat{x}$.

**Graph neural networks for node classification.** In node classification, the task is to label each node in a graph based on its attributes and connections with other nodes. GNNs operate by aggregating information from a node's neighbors, then updating the node's representation based on the aggregated information. Formally, the $k$-th layer of a GNN is designed as:

$$h_i^k = f^k(h_i^{k-1}, \text{AGG}(\{h_j^{k-1} : j \in \mathcal{N}_i\})) \in \mathbb{R}^d, \tag{3}$$

where $h_i^k \in \mathbb{R}^d$ is the representation of node $i$ at layer $k$ and $\mathcal{N}_i \subseteq \mathcal{V}$ is the set of neighbors of node $i$. Function $f^k$ is a differentiable function that updates the representation of a node based on its previous-layer representation and the aggregated information from its neighbors. This function

is typically implemented as a neural network layer (*e.g.,* a multi-layer perceptron, or an attention mechanism). AGG is also a differentiable function (*e.g.,* sum, mean, etc.) that aggregates the representations of a node's neighbors to produce a summary vector. The final representation is fed into a fully connected layer and a softmax function for class prediction.

# 4 PROPOSED METHOD

In this section, we describe our LLM-based pipeline designed for node classification on TAGs. As illustrated in Figure 1, the key idea is to leverage the LLM's explanations as informative features for a downstream GNN. To achieve this goal, our method involves three main steps: 1) LLM-based prediction and explanation generation, 2) fine-tuning an LM interpreter, and 3) training a GNN.

## 4.1 GENERATING PREDICTIONS AND EXPLANATIONS WITH LLMS

As outlined in the introduction, our approach is designed to be *LMaaS-compatible* given the scale of LLMs. This means that we aim to operate solely through API access to an LLM, using text-based input and output, without requiring fine-tuning the LLM or accessing its embeddings or logits.

In lieu of these requirements, our approach focuses on querying the LLM in an "open-ended" manner, *i.e.,* instructing the LLM to make multiple predictions and provide explanations for its decisions. By doing so, we aim to effectively extract its reasoning abilities and general knowledge in text format. These text-based outputs are then processed using an *LLM-to-LM interpreter* to create informative node features for downstream GNNs. With this objective, for each paper node $i \in \mathcal{V}$, we generate a prompt that includes the title and abstract of the paper, along with an open-ended question about the paper's topic. The specific phrasing of the question part of the prompt is tailored to the task and dataset, as shown in Table 5. The general structure of the prompt is as follows:

> **Abstract:** [paper abstract]
> **Title:** [paper title]
> **Question:** [ask the model to predict one or more class labels of the paper, ordered from most to least likely, and provide explanations for its predictions]
> **Answer:**

Querying the LLM results in a ranked prediction list and a textual explanation for each paper:

> (**Ranked Predictions**) [a ranked prediction list]
> (**Explanations**) [model-generated explanation for the predictions]

These predictions and explanations serve as supplementary text attributes for the downstream LMs and GNN models, as detailed in the subsequent section.

## 4.2 FINE-TUNING LM INTERPRETER AND NODE FEATURE EXTRACTION

**Original text and explanation features.** Our initial step involves converting both the original text, *i.e.,* title and abstract, and the LLM's explanations into fixed-length node features suitable for downstream GNN applications. Our approach is to fine-tune a smaller LM, which acts as an "interpreter" for the LLM's text explanations. The rationale behind this step is that both the LLM and LM possess distinct advantages: the LLM has greater power and more knowledge but is less flexible, while the LM has less skills but is compact enough to be fine-tuned to a specific task. Thus, the LM serves to interpret the LLM's output for the GNN, with the text explanation acting as an effective intermediate medium for communication. Then, fine-tuning the LM enables it to extract the most valuable and task-relevant features from the explanations.

Concretely, we first fine-tune pre-trained LMs as follows: let $\text{LM}_{\text{orig}}$ and $\text{LM}_{\text{expl}}$ be pre-trained LMs that take as input the original $s^{\text{orig}}$ and the explanation $s^{\text{expl}}$ text sequences, respectively. We obtain text embeddings for each source as follows:

$$h_{\text{orig}} = \text{LM}_{\text{orig}}(s^{\text{orig}}) \in \mathbb{R}^{N \times d}, \quad h_{\text{expl}} = \text{LM}_{\text{expl}}(s^{\text{expl}}) \in \mathbb{R}^{N \times d}. \tag{4}$$

We further apply a Multi-Layer Perceptron (MLP) to the output of the LMs to obtain a $N \times C$-dimensional prediction matrix representing the LM's predictions for each node (in logits):

$$y_{\text{orig}} = \text{MLP}_{\text{orig}}(h_{\text{orig}}) \in \mathbb{R}^{N \times C}, \quad y_{\text{expl}} = \text{MLP}_{\text{expl}}(h_{\text{expl}}) \in \mathbb{R}^{N \times C}. \tag{5}$$

We fine-tune these LMs and MLPs using cross-entropy loss. Finally, the text embeddings from both sources, $h_{\text{orig}}$ and $h_{\text{expl}}$, are used as enriched features for training downstream GNNs.

**Ranked prediction features.** In addition to the explanations, the LLM also provides a top-$k$ ranked prediction list for each node, which adds valuable information. To incorporate this knowledge, the top-$k$ predictions for node $i$ are first one-hot encoded as vectors $p_{i,1}, \ldots, p_{i,k} \in \mathbb{R}^C$. These vectors are subsequently concatenated into a $kC$-dimensional vector, followed by a linear transformation to produce a fixed-sized vector of length $d_P$. This process produces a prediction feature matrix as $h_{\text{pred}} \in \mathbb{R}^{N \times d_P}$ across all nodes.

In summary, we denote our features as $h_{\text{TAPE}} = \{h_{\text{orig}}, h_{\text{expl}}, h_{\text{pred}}\}$, where "TAPE" stands for Title, Abstract, Prediction and Explanation for each node. Importantly, our framework requires these features to remain frozen during downstream GNN training, ensuring that the LM and LLM do not participate in the GNN training process. This characteristic significantly enhances ease-of-use, modularity, and efficiency compared to approaches like GLEM, which involve an expensive iterative LM-GNN training process. As a result, we achieve a substantial speedup over GLEM, *e.g.,* a $2.88\times$ speedup on `ogbn-arxiv` even when utilizing the same backbone LM and GNN.

### 4.3 GNN TRAINING ON ENRICHED FEATURES

Our final step is to train a GNN using the $h_{\text{TAPE}}$ features. We aim to achieve this without increasing the memory requirements of the GNN or making any changes to its architecture. To accomplish this, we use an ensemble approach, as a simple and effective way of combining the features. Specifically, we independently train GNN models $f_{\text{orig}}$, $f_{\text{expl}}$, and $f_{\text{pred}}$ on the features $h_{\text{orig}}$, $h_{\text{expl}}$, and $h_{\text{pred}}$, respectively, to predict the ground truth node labels:

$$\hat{y}_{\text{orig/expl/pred}} = f_{\text{orig/expl/pred}}(h_{\text{orig/expl/pred}}, A) \in \mathbb{R}^{N \times C}. \tag{6}$$

We then fuse these predictions by taking their average:

$$\hat{y} = \text{mean}(\hat{y}_{\text{orig}}, \hat{y}_{\text{expl}}, \hat{y}_{\text{pred}}) \in \mathbb{R}^{N \times C}. \tag{7}$$

Each of the three models performs well individually as shown in Table 3, which validates the effectiveness of simple averaging. This strategy enables us to capture complementary information from diverse input sources, ultimately enhancing the overall model's performance.

### 4.4 THEORETICAL ANALYSIS

In this section, we aim to demonstrate that explanations generated by an LLM can be valuable features for a smaller LM. Specifically, the explanations $E$ are helpful if they possess *fidelity* in describing the LLM's reasoning; and the LLM is *non-redundant*, utilizing information not used by the smaller LM. Let $E$ be the textual explanations generated by an LLM; $Z_L$ and $Z$ are embeddings from the LLM and smaller LM respectively, $y$ is the target and $H(\cdot|\cdot)$ is the conditional entropy. The detailed proof is in Appendix A.

**Theorem.** Given the following conditions 1) *Fidelity*: $E$ is a good proxy for $Z_L$ such that $H(Z_l|E) = \epsilon$, with $\epsilon > 0$, 2) *Non-redundancy*: $Z_L$ contains information not present in $Z$, expressed as $H(y|Z, Z_L) = H(y|Z) - \epsilon'$, with $\epsilon' > \epsilon$. Then it follows that $H(y|Z, E) < H(y|Z)$.

## 5 EXPERIMENTS

We evaluate TAPE on five TAG datasets: `Cora` (McCallum et al., 2000), `PubMed` (Sen et al., 2008), `ogbn-arxiv`, `ogbn-products` (Hu et al., 2020a), and `tape-arxiv23`. For `Cora` and `PubMed`, raw text data of the articles is unavailable in common graph libraries such as PyG and DGL. Hence, we collected and formatted the missing text data for these datasets in TAG format. Additionally, given the popularity of these datasets, their TAG version will be released publicly for reproducibility and new research projects. For `ogbn-products`, given its substantial scale of 2 million nodes and 61 million edges and considering our academic resource budget, we conducted experiments on a subgraph sample. Details can be found in Appendix G.

## 5.1 MAIN RESULTS

Table 1: Node classification accuracy for the `Cora`, `PubMed`, `ogbn-arxiv`, `ogbn-products` and `tape-arxiv23` datasets. $G \uparrow$ denotes the improvements of our approach over the same GNN trained on shallow features $h_{\text{shallow}}$; $L \uparrow$ denotes the improvements of our approach over $\text{LM}_{\text{finetune}}$. The results are averaged over four runs with different seeds, and the best results are **in bold**.

| Dataset | Method | GNN | | | LM | | | Ours |
|---|---|---|---|---|---|---|---|---|
| | | $h_{\text{shallow}}$ | $h_{\text{GIANT}}$ | $G \uparrow$ | LLM | $\text{LM}_{\text{finetune}}$ | $L \uparrow$ | $h_{\text{TAPE}}$ |
| `Cora` | MLP | $0.6388 \pm 0.0213$ | $0.7196 \pm 0.0000$ | 37.41% | 0.6769 | $0.7606 \pm 0.0378$ | 13.35% | $0.8778 \pm 0.0485$ |
| | GCN | $0.8911 \pm 0.0015$ | $0.8423 \pm 0.0053$ | 2.33% | 0.6769 | $0.7606 \pm 0.0378$ | 16.59% | $0.9119 \pm 0.0158$ |
| | SAGE | $0.8824 \pm 0.0009$ | $0.8455 \pm 0.0028$ | 5.28% | 0.6769 | $0.7606 \pm 0.0378$ | 18.13% | $\mathbf{0.9290 \pm 0.0307}$ |
| | RevGAT | $0.8911 \pm 0.0000$ | $0.8353 \pm 0.0038$ | 4.14% | 0.6769 | $0.7606 \pm 0.0378$ | 18.04% | $0.9280 \pm 0.0275$ |
| `PubMed` | MLP | $0.8635 \pm 0.0032$ | $0.8175 \pm 0.0059$ | 10.77% | 0.9342 | $0.9494 \pm 0.0046$ | 0.75% | $0.9565 \pm 0.0060$ |
| | GCN | $0.8031 \pm 0.0425$ | $0.8419 \pm 0.0050$ | 17.43% | 0.9342 | $0.9494 \pm 0.0046$ | -0.66% | $0.9431 \pm 0.0043$ |
| | SAGE | $0.8881 \pm 0.0002$ | $0.8372 \pm 0.0082$ | 8.30% | 0.9342 | $0.9494 \pm 0.0046$ | 1.31% | $\mathbf{0.9618 \pm 0.0053}$ |
| | RevGAT | $0.8850 \pm 0.0005$ | $0.8502 \pm 0.0048$ | 8.52% | 0.9342 | $0.9494 \pm 0.0046$ | 1.15% | $0.9604 \pm 0.0047$ |
| `ogbn-arxiv` | MLP | $0.5336 \pm 0.0038$ | $0.7308 \pm 0.0006$ | 42.19% | 0.7350 | $0.7361 \pm 0.0004$ | 3.07% | $0.7587 \pm 0.0015$ |
| | GCN | $0.7182 \pm 0.0027$ | $0.7329 \pm 0.0010$ | 4.71% | 0.7350 | $0.7361 \pm 0.0004$ | 2.16% | $0.7520 \pm 0.0003$ |
| | SAGE | $0.7171 \pm 0.0017$ | $0.7435 \pm 0.0014$ | 6.98% | 0.7350 | $0.7361 \pm 0.0004$ | 4.22% | $0.7672 \pm 0.0007$ |
| | RevGAT | $0.7083 \pm 0.0017$ | $0.7590 \pm 0.0019$ | 9.42% | 0.7350 | $0.7361 \pm 0.0004$ | 5.28% | $\mathbf{0.7750 \pm 0.0012}$ |
| `ogbn-products` | MLP | $0.5385 \pm 0.0017$ | $0.6125 \pm 0.0078$ | 46.3% | 0.7440 | $0.7297 \pm 0.0023$ | 7.96% | $0.7878 \pm 0.0082$ |
| | GCN | $0.7052 \pm 0.0051$ | $0.6977 \pm 0.0042$ | 13.39% | 0.7440 | $0.7297 \pm 0.0023$ | 9.58% | $0.7996 \pm 0.0041$ |
| | SAGE | $0.6913 \pm 0.0026$ | $0.6869 \pm 0.0119$ | 17.71% | 0.7440 | $0.7297 \pm 0.0023$ | 11.51% | $0.8137 \pm 0.0043$ |
| | RevGAT | $0.6964 \pm 0.0017$ | $0.7189 \pm 0.0030$ | 18.24% | 0.7440 | $0.7297 \pm 0.0023$ | 12.84% | $\mathbf{0.8234 \pm 0.0036}$ |
| `tape-arxiv23` | MLP | $0.6202 \pm 0.0064$ | $0.5574 \pm 0.0032$ | 35.20% | 0.7356 | $0.7358 \pm 0.0006$ | 12.25% | $0.8385 \pm 0.0246$ |
| | GCN | $0.6341 \pm 0.0062$ | $0.5672 \pm 0.0061$ | 27.42% | 0.7356 | $0.7358 \pm 0.0006$ | 8.94% | $0.8080 \pm 0.0215$ |
| | SAGE | $0.6430 \pm 0.0037$ | $0.5665 \pm 0.0032$ | 30.45% | 0.7356 | $0.7358 \pm 0.0006$ | 12.28% | $0.8388 \pm 0.0264$ |
| | RevGAT | $0.6563 \pm 0.0062$ | $0.5834 \pm 0.0038$ | 28.34% | 0.7356 | $0.7358 \pm 0.0006$ | 12.64% | $\mathbf{0.8423 \pm 0.0256}$ |

We conduct a comprehensive evaluation of our proposed TAPE method by comparing with existing GNN- and LM-based methods, with the results summarized in Table 1. For GNN comparisons, we consider three widely utilized architectures: GCN (Kipf & Welling, 2016), GraphSAGE (Sun et al., 2021), and RevGAT (Li et al., 2021) along with a basic MLP baseline that operates independently off graph-related information. We explore three types of node features: 1) shallow features (detailed in Table 9), denoted as $h_{\text{shallow}}$, 2) GIANT features (Chien et al., 2021) $h_{\text{GIANT}}$, and 3) our proposed features $h_{\text{TAPE}}$, comprising $h_{\text{orig}}$, $h_{\text{expl}}$, and $h_{\text{pred}}$. For LM-based methods, we investigate two approaches: 1) fine-tuning DeBERTa on labeled nodes, denoted as $\text{LM}_{\text{finetune}}$, and 2) using zero-shot ChatGPT (gpt-3.5-turbo) with the same prompts as our approach, denoted as LLM.

Our approach consistently outperforms other methods on all datasets and across all models, demonstrating its effectiveness in enhancing TAG representation learning. Among GNN-based methods, shallow features (*i.e.,* $h_{\text{shallow}}$) yields subpar performance, while LM-based features (*i.e.,* $h_{\text{GIANT}}$) improves results. In the case of LMs, fine-tuned LMs (*i.e.,* $\text{LM}_{\text{finetune}}$) also perform well. Our proposed novel features, leveraging the power of the LLM, further enhance the results.

Additionally, we expanded our experimentation to include the open-source Llama2 (Touvron et al., 2023), demonstrating the feasibility of a cost-effective (free) alternative, as shown in Table 4. Furthermore, to address the potential label leakage concern in LLM, we took the initiative to construct a novel dataset, namely `tape-arxiv23`, comprising papers published in 2023 or later – well beyond the knowledge cutoff for GPT-3.5. The results clearly illustrate strong generalization capabilities: while the LLM achieves 73.56% accuracy, our approach outperforms it with 84.23%.

## 5.2 SCALABILITY

Our proposed method surpasses not only pure LMs and shallow embedding pipelines but also the LM-based pipelines on the `ogbn-arxiv` dataset, achieving a superior balance between accuracy and training time, as illustrated in Figure 2. Specifically, our method achieved significantly higher accuracy than the SOTA GLEM (Zhao et al., 2022) method while utilizing the same LM and GNN models. Furthermore, our approach requires only $2.88\times$ less computation time. These efficiency improvements are attributed to our decoupled training approach for LMs and GNNs, avoiding the iterative (*i.e.,* multi-stage) approach used in GLEM. Moreover, unlike the iterative approach, our model allows for parallelizing the training of $\text{LM}_{\text{orig}}$ and $\text{LM}_{\text{expl}}$, further reducing overall training time when performed simultaneously.

Table 2: Experiments on `ogbn-arxiv` dataset with DeBERTa-base (He et al., 2021) as LM backbone and RevGAT (Li et al., 2021) as GNN backbone for comparison of different training paradigms of fusing LMs and GNNs, including our proposed method and the state-of-the-art GLEM method (Zhao et al., 2022). The validation and test accuracy, number of parameters, maximum batch size (Max bsz.), and total training time on 4 NVIDIA RTX A5000 24GB GPUs are reported.

| Method | Val acc. | Test acc. | Params. | Max bsz. | Total time |
|---|---|---|---|---|---|
| $\text{LM}_{orig}$ | $0.7503 \pm 0.0008$ | $0.7361 \pm 0.0004$ | 139,223,080 | 36 | 1.73h |
| GNN-$h_{\text{shallow}}$ | $0.7144 \pm 0.0021$ | $0.7083 \pm 0.0017$ | 427,728 | all nodes | 1.80min |
| GLEM-G-Step | $0.7761 \pm 0.0005$ | $0.7657 \pm 0.0029$ | 1,837,136 | all nodes | 9.18h |
| GLEM-L-Step | $0.7548 \pm 0.0039$ | $0.7495 \pm 0.0037$ | 138,632,488 | 36 | |
| TAPE-$\text{LM}_{orig}$-Step | $0.7503 \pm 0.0008$ | $0.7361 \pm 0.0004$ | 139,223,080 | 36 | 1.73h |
| TAPE-$\text{LM}_{expl}$-Step | $0.7506 \pm 0.0008$ | $0.7432 \pm 0.0012$ | 139,223,080 | 36 | 1.40h |
| TAPE-GNN-$h_{\text{TAPE}}$-Step | $0.7785 \pm 0.0016$ | $0.7750 \pm 0.0012$ | 1,837,136 | all nodes | 3.76min |

Table 3: Ablation study on the `ogbn-arxiv` dataset, showing the effects of different node features on the performance. Node features include the original text attributes ($h_{\text{orig}}$), the explanations ($h_{\text{expl}}$ and predicted $h_{\text{pred}}$) generated by LLM, and the proposed method ($h_{\text{TAPE}}$). Results are averaged over 4 runs with 4 different seeds. The best results are **in bold**.

| Method | | $h_{\text{orig}}$ | $h_{\text{expl}}$ | $h_{\text{pred}}$ | $h_{\text{TAPE}}$ |
|---|---|---|---|---|---|
| GCN | val | $0.7624 \pm 0.0007$ | $0.7577 \pm 0.0008$ | $0.7531 \pm 0.0006$ | $0.7642 \pm 0.0003$ |
| | test | $0.7498 \pm 0.0018$ | $0.7460 \pm 0.0013$ | $0.7400 \pm 0.0007$ | $\textbf{0.7520} \pm \textbf{0.0003}$ |
| SAGE | val | $0.7594 \pm 0.0012$ | $0.7631 \pm 0.0016$ | $0.7612 \pm 0.0010$ | $0.7768 \pm 0.0016$ |
| | test | $0.7420 \pm 0.0018$ | $0.7535 \pm 0.0023$ | $0.7524 \pm 0.0015$ | $\textbf{0.7672} \pm \textbf{0.0007}$ |
| RevGAT | val | $0.7588 \pm 0.0021$ | $0.7568 \pm 0.0027$ | $0.7550 \pm 0.0015$ | $0.7785 \pm 0.0016$ |
| | test | $0.7504 \pm 0.0020$ | $0.7529 \pm 0.0052$ | $0.7519 \pm 0.0031$ | $\textbf{0.7750} \pm \textbf{0.0012}$ |

## 5.3 ABLATION STUDY

We perform an ablation study on the `ogbn-arxiv` dataset (Hu et al., 2020a) to evaluate the relevance of each module within our framework. The results are summarized in Table 3 and Figure 4. Across all methods and for both the validation and test sets, our proposed method consistently outperforms the other settings. This underscores the value of incorporating explanations and predictions into node embeddings.

We provide time analysis and cost estimation in Appendix B, detail `tape-arxiv23` dataset collection in Appendix C, use open-sourced llama as the LLM in Appendix D, include a case study in Appendix E, discuss prompt design in Appendix F, examine LM finetuning effects in Appendix I, explore the impact of various LMs in Appendix J, and analyze memory usage in Appendix K.

## 6 CONCLUSION

Given the increasing importance of integrating text and relationships, coupled with the emergence of LLMs, we foresee that TAG tasks will attract even more attention in the coming years. The convergence of LLMs and GNNs presents new opportunities for both research and industrial applications. As a pioneering work in this field, we believe that our contribution will serve as a strong baseline for future studies in this domain.

**Limitation and future work.** An inherent limitation of our approach lies in the requirement for customized prompts for each dataset. Currently, we rely on manually crafted prompts, which may not be optimal for the node classification task for every dataset. The efficacy of these prompts may fluctuate depending on the specific characteristics of the dataset and the specific task at hand. Future work can focus on automating the prompt generation process, exploring alternative prompt designs, and addressing the challenges of dynamic and evolving TAGs.

ACKNOWLEDGMENT

Bryan Hooi is supported by the Ministry of Education, Singapore, under the Academic Research Fund Tier 1 (FY2023) (Grant A-8001996-00-00) and Xavier Bresson is supported by NUS Grant ID R-252-000-B97-133. The authors would like to express their gratitude to the reviewers for their feedback, which has improved the clarity and contribution of the paper.

REPRODUCIBILITY STATEMENT

In this statement, we provide references to the relevant sections and materials that will assist readers and researchers in replicating our results.

**Theorem.** For a comprehensive understanding of the theorem presented in Section 4.4, please refer to Appendix A for a detailed proof.

**Dataset description.** We summarize all datasets used in our study in Appendix G, providing information on their sources and any necessary preprocessing steps. Additionally, for the newly introduced `tape-arxiv23` dataset, we offer a detailed description of the data collection and processing steps in Appendix C.

**Open access to codes, datasets, trained models, and enriched features.** Our source code can be accessed at the following url: `https://github.com/XiaoxinHe/TAPE`. Within this repository, we provide a script with step-by-step instructions on how to replicate the main results presented in our paper. Additionally, we offer download links for the `Cora` and `PubMed` datasets in TAG form, along with the new dataset `tape-arxiv23`. These datasets can serve as valuable resources for the NLP and GNN research community. Furthermore, this repository includes the checkpoints for all trained models (.ckpt) and the TAPE features (.emb) used in our project, making it easy for researchers focusing on downstream GNN tasks to access enriched features.

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

## A    THEORETICAL ANALYSIS

In this section, we aim to demonstrate that explanations generated by an LLM can provide valuable features for another model (such as a smaller LM). This is true under two key conditions:

1. *Fidelity:* The explanations effectively represent LLM's reasoning over the raw text, containing most of the information from the LLM's hidden state.
2. *Non-redundancy:* The LLM possesses unique knowledge not captured by another model.

We formulate our theorem as follows:

**Theorem 1.** *Given the following conditions:*

*1) Fidelity: $E$ is a good proxy for $Z_L$ such that*

$$H(Z_l|E) = \epsilon, \quad \epsilon > 0 \tag{8}$$

*2) Non-redundancy: $Z_L$ contains information not present in $Z$, expressed as*

$$H(y|Z, Z_L) = H(y|Z) - \epsilon', \quad \epsilon' > \epsilon \tag{9}$$

*Then, it follows that:*

$$H(y|Z, E) < H(y|Z) \tag{10}$$

*where $E$ is textual explanations generated by an LLM, $Z_L$ is the vectorial representation of the raw text modeled by the LLM, $Z$ is the vectorial representation of the raw text modeled by the other model, $y$ is the target and $H(\cdot|\cdot)$ is the conditional entropy.*

*Proof.* We aim to demonstrate that the conditional entropy of $y$ given both $Z$ and $E$, denoted as $H(y|Z, E)$, is less than the conditional entropy of $y$ given only $Z$, denoted as $H(y|Z)$.

Starting with:

$$H(y|Z, E) \tag{11}$$

We apply the properties of entropy to decompose this expression into two components:

$$H(y|Z, E) = H(y|Z, Z_L, E) + I(y; Z_L|Z, E) \tag{12}$$

Now, we utilize the following upper bound of conditional mutual information:

$$I(y; Z_L|Z, E) = H(Z_L|Z, E) - H(Z_L|y, Z, E) \tag{13}$$
$$\leq H(Z_L|Z, E) \tag{14}$$

where the first line follows from the definition of mutual information, and the second line follows from the nonnegativity of conditional entropy.

Substituting equation 14 into equation 12, we rewrite the conditional entropy as:

$$H(y|Z, E) \leq H(y|Z, Z_L, E) + H(Z_L|Z, E) \tag{15}$$

Since conditional entropy increases when conditioning on fewer variables, we further have:

$$H(y|Z, Z_L, E) + H(Z_L|Z, E) \leq H(y|Z, Z_L) + H(Z_L|E) \tag{16}$$

Applying the "Fidelity" and "Non-redundancy" conditions:

$$H(y|Z, Z_L) + H(Z_L|E) \leq H(y|Z) - \epsilon' + \epsilon \tag{17}$$

Finally, as $\epsilon' > \epsilon$, we have:

$$H(y|Z) - \epsilon' + \epsilon < H(y|Z) \tag{18}$$

Consequently, we have proven that:

$$H(y|Z, E) < H(y|Z) \tag{19}$$

This completes the proof. □

## B    TIME ANALYSIS AND MONEY ESTIMATION

Our primary dataset, `ogbn-arxiv`, with 169,343 nodes and 1,166,243 edges, serves as a representative case for our approach. On average, our input sequences consist of approximately 285 tokens, while the output sequences comprise around 164 tokens. For the ChatGPT-3.5 Turbo API, priced at \$0.0015 per 1,000 input tokens and \$0.002 per 1,000 output tokens, with a token per minute rate limit of 90,000, the monetary estimation for `ogbn-arxiv` is as follows:

$$Cost = ((285 \times 0.0015)/1000 + (164 \times 0.002)/1000) \times 169,343 \approx 128\,USD \tag{20}$$

Considering the token rate limit, we estimate the deployment time as follows:

$$Time = 169,343/(90,000/285) \approx 536min \approx 9h \tag{21}$$

**Cost-Effective Alternatives.** Additionally, we have explored cost-effective alternatives, such as leveraging open-source LLMs like llama2. The use of llama2 is entirely free, and the querying process to llama2-13b-chat takes approximately 16 hours when utilizing 4 A5000 GPUs.

**Efficiency through Single Query and Reuse.** Our method requires only one query to the LLM, with predictions and explanations stored for subsequent use. This not only enhances efficiency but also minimizes the number of API calls, contributing to cost-effectiveness. We also release the gpt responses for public use.

## C    ADDRESSING LABEL LEAKAGE CONCERNS WITH A NEW DATASET

GPT-3.5's training data might include certain arXiv papers, given its comprehensive ingestion of textual content from the internet. However, the precise composition of these arXiv papers within GPT-3.5's training remains undisclosed, rendering it infeasible to definitively identify their inclusion. It is essential to emphasize that the challenge of label leakage is widespread and affects various language model benchmarks, such as the prominent BIG-bench (Srivastava et al., 2022) and TruthfulQA (Lin et al., 2021).

To address this concern, we created a novel dataset `tape-arxiv23` for our experiments. We made sure that this dataset only included papers published in 2023 or later, which is well beyond the knowledge cutoff for GPT-3.5, as it was launched in November 2022. The creation of this new dataset was meticulously executed. We collected all cs.ArXiv papers published from January 2023 to September 2023 from the arXiv daily repository [2]. We then utilized the Semantic Scholar API [3] to retrieve citation relationships. This process yielded a comprehensive graph containing 46,198 papers and 78,548 connections. Our codes to collect and build the dataset is available at: `https://github.com/XiaoxinHe/tape_arxiv_2023`.

## D    LLAMA AS A COST-EFFICIENT ALTERNATIVE

We extend out experiment to the open-source LLM "llama-2-13b-chat" (llama for short), which demonstrates the feasibility of a cost-effective (free) alternative, see Table 4.

It is worth noting that although llama exhibits a lower performance compared to GPT-3.5 in terms of both zero-shot accuracy and explanation quality, our pipeline still maintains its robust performance. As an illustration, we achieved an accuracy of 76.19% on the `ogbn-arxiv` dataset using llama, slightly below the 77.50% achieved with GPT-3.5. We attribute this impressive level of generalization to the complementary nature of the explanations themselves, which serve as a rich source of semantic information supplementing the original text such as title and abstract.

---

[2] `https://arxiv.org/`
[3] `https://www.semanticscholar.org/product/api`

Table 4: Node classification accuracy for the `Cora`, `PubMed` and `ogbn-arxiv` datasets.

| Dataset | Method | llama2-13b-chat | | | GPT3.5 | | |
|---|---|---|---|---|---|---|---|
| | | LLM | $LM_{finetune}$ | $h_{TAPE}$ | LLM | $LM_{finetune}$ | $h_{TAPE}$ |
| `Cora` | GCN | 0.5746 | 0.6845 ± 0.0194 | 0.9045 ± 0.0231 | 0.6769 | 0.7606 ± 0.0378 | 0.9119 ± 0.0158 |
| | SAGE | 0.5746 | 0.6845 ± 0.0194 | 0.9170 ± 0.0337 | 0.6769 | 0.7606 ± 0.0378 | 0.9290 ± 0.0307 |
| | RevGAT | 0.5746 | 0.6845 ± 0.0194 | 0.9313 ± 0.0237 | 0.6769 | 0.7606 ± 0.0378 | 0.9280 ± 0.0275 |
| `PubMed` | GCN | 0.3958 | 0.9121 ± 0.0026 | 0.9362 ± 0.0050 | 0.9342 | 0.9494 ± 0.0046 | 0.9431 ± 0.0043 |
| | SAGE | 0.3958 | 0.9121 ± 0.0026 | 0.9581 ± 0.0073 | 0.9342 | 0.9494 ± 0.0046 | 0.9618 ± 0.0053 |
| | RevGAT | 0.3958 | 0.9121 ± 0.0026 | 0.9561 ± 0.0068 | 0.9342 | 0.9494 ± 0.0046 | 0.9604 ± 0.0047 |
| `ogbn-arxiv` | GCN | 0.4423 | 0.6941 ± 0.0020 | 0.7418 ± 0.0031 | 0.7350 | 0.7361 ± 0.0004 | 0.7520 ± 0.0003 |
| | SAGE | 0.4423 | 0.6941 ± 0.0020 | 0.7536 ± 0.0028 | 0.7350 | 0.7361 ± 0.0004 | 0.7672 ± 0.0007 |
| | RevGAT | 0.4423 | 0.6941 ± 0.0020 | 0.7619 ± 0.0027 | 0.7350 | 0.7361 ± 0.0004 | 0.7750 ± 0.0012 |
| `tape-arxiv23` | GCN | 0.4452 | 0.7677 ± 0.0042 | 0.8045 ± 0.0264 | 0.7356 | 0.7832 ± 0.0052 | 0.8080 ± 0.0215 |
| | SAGE | 0.4452 | 0.7677 ± 0.0042 | 0.8378 ± 0.0302 | 0.7356 | 0.7832 ± 0.0052 | 0.8388 ± 0.0264 |
| | RevGAT | 0.4452 | 0.7677 ± 0.0042 | 0.8407 ± 0.0308 | 0.7356 | 0.7832 ± 0.0052 | 0.8423 ± 0.0256 |

# E  CASE STUDY

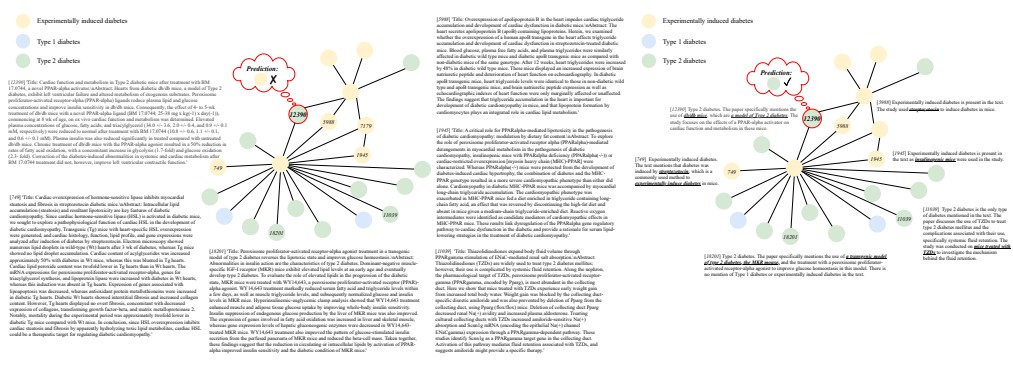

(a) Original text attributes as features.    (b) Explanations as features.

Figure 3: Case study comparing features for node classification on the `PubMed` dataset: (a) Original text attributes and (b) Explanations generated by LLMs. The GNN model trained with (b) accurately predicts the label for node 12390 (type 2 diabetes), while the model trained with (a) predicts the incorrect label (experimentally induced diabetes). This improvement can be attributed to the concise and focused nature of LLM-generated explanations, as well as their reasoning ability and utilization of external knowledge.

To investigate the impact of using explanations as features in improving node classification on TAGs, we conduct an analysis on predicted samples from the `PubMed` dataset. Figure 3 presents a case where the GNN model trained with original text attributes as features incorrectly predicts the label for node 12390 (as experimentally induced diabetes), while the model trained with explanations generated by LLMs as features correctly predicts the label (as type 2 diabetes).

This improvement can be attributed to two main factors. Firstly, compared to the original text attributes, which consist of the title and abstract text, the explanations generated by the LLM are more concise and focused. This aids the subsequent LM in generating node embeddings that capture the essential semantics without the need to compress an excessive amount of information into a fixed-length representation. Secondly, LLMs possess reasoning capabilities and the ability to leverage general knowledge, which prove crucial in achieving accurate predictions. For instance, the explanations generated by LLMs explicitly link type 2 diabetes to MKR mice and db/db mice (which are common animal models of type 2 diabetes), as well as the insulinopenic mice / streptozotocin to experimentally induced diabetes. This knowledge is either absent or only implicitly specified in the original text attributes.

# F   PROMPT DESIGN

Table 5 outlines the prompts used for various datasets. Each prompt includes the abstract and title of the paper, followed by a task-specific question. The question is formulated to query the model about a particular aspect of the paper and request an explanation for the prediction. The answer section is left blank for the model to fill in. Generally, our analysis finds that the current instructions allow the LLM to produce output that conforms well to the expected format without significant deviations, allowing the answers to be straightforwardly extracted from the text output of the LLM.

Table 5: Prompts used in this work to query the LLM.

| Dataset | Prompt |
| --- | --- |
| Cora | Abstract: <abstract text> \n Title: <title text> \n Question: Which of the following sub-categories of AI does this paper belong to: Case Based, Genetic Algorithms, Neural Networks, Probabilistic Methods, Reinforcement Learning, Rule Learning, Theory? If multiple options apply, provide a comma-separated list ordered from most to least related, then for each choice you gave, explain how it is present in the text. \n \n Answer: |
| Pubmed | Abstract: <abstract text> \n Title: <title text> \n Question: Does the paper involve any cases of Type 1 diabetes, Type 2 diabetes, or Experimentally induced diabetes? Please give one or more answers of either Type 1 diabetes, Type 2 diabetes, or Experimentally induced diabetes; if multiple options apply, provide a comma-separated list ordered from most to least related, then for each choice you gave, give a detailed explanation with quotes from the text explaining why it is related to the chosen option. \n \n Answer: |
| ogbn-arxiv | Abstract: <abstract text> \n Title: <title text> \n Question: Which arXiv CS sub-category does this paper belong to? Give 5 likely arXiv CS sub-categories as a comma-separated list ordered from most to least likely, in the form "cs.XX", and provide your reasoning. \n \n Answer: |
| ogbn-products | Product description: <product description> \n Question: Which of the following category does this product belong to: 1) Home & Kitchen, 2) Health & Personal Care, 3) Beauty, 4) Sports & Outdoors, 5) Books, 6) Patio, Lawn & Garden, 7) Toys & Games, 8) CDs & Vinyl, 9) Cell Phones & Accessories, 10) Grocery & Gourmet Food, 11) Arts, Crafts & Sewing, 12) Clothing, Shoes & Jewelry, 13) Electronics, 14) Movies & TV, 15) Software, 16) Video Games, 17) Automotive, 18) Pet Supplies, 19) Office Products, 20) Industrial & Scientific, 21) Musical Instruments, 22) Tools & Home Improvement, 23) Magazine Subscriptions, 24) Baby Products, 25) NAN, 26) Appliances, 27) Kitchen & Dining, 28) Collectibles & Fine Art, 29) All Beauty, 30) Luxury Beauty, 31) Amazon Fashion, 32) Computers, 33) All Electronics, 34) Purchase Circles, 35) MP3 Players & Accessories, 36) Gift Cards, 37) Office & School Supplies, 38) Home Improvement, 39) Camera & Photo, 40) GPS & Navigation, 41) Digital Music, 42) Car Electronics, 43) Baby, 44) Kindle Store, 45) Kindle Apps, 46) Furniture & Decor? Give 5 likely categories as a comma-separated list ordered from most to least likely, and provide your reasoning. \n \n Answer: |
| tape-arxiv23 | Abstract: <abstract text> \n Title: <title text> \n Question: Which arXiv CS sub-category does this paper belong to? Give 5 likely arXiv CS sub-categories as a comma-separated list ordered from most to least likely, in the form "cs.XX", and provide your reasoning. \n \n Answer: |

**Exploring Prompt Variations.**   We have extensively explored the influence of various prompts on the ogbn-arxiv dataset, as outlined in Table 6 and Table 7.

Table 6 indicates that, generally, most prompts yield similar performance. However, a minor performance improvement is observed when the title is positioned after the abstract. This finding aligns with the principle suggested by Zhao et al. (2021) that placing more critical information later in the prompt can be beneficial.

Further analysis presented in Table 7 demonstrates a positive correlation between the LLM's zero-shot accuracy and the overall accuracy of our method, implying that higher zero-shot prediction scores lead to enhanced TAPE accuracy. Despite the variation in prompt designs, our methodology

consistently achieves similar accuracy levels, ranging from 0.7660 to 0.7750 with the RevGAT as the GNN backbone. This consistency underscores the robustness of our proposed TAPE to different prompt configurations.

Table 6: Prompts used for our experiments studying the effect of different prompts. Most prompts have similar performance.

| Description | Prompt | Accuracy |
|---|---|---|
| Default prompt | Abstract: <abstract text> \n Title: <title text> \n Question: Which arXiv CS sub-category does this paper belong to? Give 5 likely arXiv CS sub-categories as a comma-separated list ordered from most to least likely, in the form "cs.XX", and provide your reasoning. \n \n Answer: | 0.720 |
| Title first | Title: <title text> \n Abstract: <abstract text> \n Question: Which arXiv CS sub-category does this paper belong to? Give 5 likely arXiv CS sub-categories as a comma-separated list ordered from most to least likely, in the form "cs.XX", and provide your reasoning. \n \n Answer: | 0.695 |
| Focus on text content | Title: <title text> \n Abstract: <abstract text> \n Question: Which arXiv CS sub-category does this paper belong to? Give 5 likely arXiv CS sub-categories as a comma-separated list ordered from most to least likely, in the form "cs.XX". Focus only on content in the actual text and avoid making false associations. Then provide your reasoning. | 0.695 |
| Chain of thought prompt | Title: <title text> \n Abstract: <abstract text> \n Question: Which arXiv CS sub-category does this paper belong to? Give 5 likely arXiv CS sub-categories as a comma-separated list ordered from most to least likely, in the form "cs.XX". Please think about the categorization in a step by step manner and avoid making false associations. Then provide your reasoning. | 0.705 |

Table 7: Study of the robustness of prompt on `ogbn-arxiv` dataset.

| | LLM (zero-shot) | TAPE (GCN) | TAPE (SAGE) | TAPE (RevGAT) |
|---|---|---|---|---|
| Default prompt | 0.720 | $0.7520 \pm 0.0003$ | $0.7672 \pm 0.0007$ | $0.7750 \pm 0.0012$ |
| Focus on text content | 0.695 | $0.7425 \pm 0.0021$ | $0.7598 \pm 0.0006$ | $0.7660 \pm 0.0017$ |
| Chain of thought prompt | 0.705 | $0.7424 \pm 0.0019$ | $0.7597 \pm 0.0034$ | $0.7667 \pm 0.0028$ |

## G  DATASET

We conduct experiments on five TAGs – `Cora` (McCallum et al., 2000), `PubMed` (Sen et al., 2008), `ogbn-arxiv`, `ogbn-products` (Hu et al., 2020a), and `tape-arxiv23`. For `Cora` and `PubMed`, we collected the raw text data since they are not available in common repositories like PyG and DGL. For `ogbn-products`, given its substantial scale of 2 million nodes and 61 million edges, we have employed a node sampling strategy to obtain a subgraph containing 54k nodes and 74k edges. Additionally, we introduced the `tape-arxiv23` citation graph dataset, extending beyond the knowledge cutoff of GPT-3. This dataset serves as a valuable resource for the research community. Table 8 provides a summary of the dataset statistics.

### G.1  DATASET DESCRIPTION

**Cora (McCallum et al., 2000).** The `Cora` dataset comprises 2,708 scientific publications classified into one of seven classes – case based, genetic algorithms, neural networks, probabilistic methods, reinforcement learning, rule learning, and theory, with a citation network consisting of 5,429 links. The papers were selected in a way such that in the final corpus every paper cites or is cited by at least one other paper.

Table 8: Statistics of the TAG datasets

| Dataset | #Nodes | #Edges | Task | Metric | Augmentation |
|---|---|---|---|---|---|
| Cora | 2,708 | 5,429 | 7-class classif. | Accuracy | ✓ |
| Pubmed | 19,717 | 44,338 | 3-class classif. | Accuracy | ✓ |
| ogbn-arxiv | 169,343 | 1,166,243 | 40-class classif. | Accuracy | |
| ogbn-products (subset) | 54,025 | 74,420 | 47-class classif. | Accuracy | |
| tape-arxiv23 | 46,198 | 78,548 | 40-class-classif. | Accuracy | ✓ |

**PubMed (Sen et al., 2008).** The Pubmed dataset consists of 19,717 scientific publications from PubMed database pertaining to diabetes classified into one of three classes – Experimental induced diabetes, Type 1 diabetes, and Type 2 diabetes. The citation network consists of 44,338 links.

**ogbn-arxiv (Hu et al., 2020a).** The `ogbn-arxiv` dataset is a directed graph that represents the citation network between all computer science arXiv papers indexed by MAG (Wang et al., 2020). Each node is an arXiv paper, and each directed edge indicates that one paper cites another one. The task is to predict the 40 subject areas of arXiv CS papers, *e.g.,*, cs.AI, cs.LG, and cs.OS, which are manually determined (*i.e.,* labeled) by the paper's authors and arXiv moderators.

**ogbn-products (Hu et al., 2020a).** The `ogbn-products` dataset represents an Amazon product co-purchasing network, with product descriptions as raw text. Nodes represent products sold in Amazon, and edges between two products indicate that the products are purchased together. The task is to predict the category of a product in a multi-class classification setup, where the 47 top-level categories are used for target labels.

**tape-arxiv23**. The `tape-arxiv23` dataset is a directed graph that represents the citation network between all computer science arXiv papers published in 2023 or later. Similar to `ogbn-arxiv`, each node is an arXiv paper, and each directed edge indicates that one paper cites another one. The task is to predict the 40 subject areas of arXiv CS papers, *e.g.,*, cs.AI, cs.LG, and cs.OS, which are manually determined (*i.e.,* labeled) by the paper's authors and arXiv moderators.

### G.2 DATASET SPLITS AND RANDOM SEEDS

In our experiments, we adhered to specific dataset splits and employed random seeds for reproducibility. For the `ogbn-arxiv` and `ogbn-products` dataset, we adopted the standard train/validation/test split provided by OGB (Hu et al., 2020a). As for the `Cora`, `PubMed` datasets, and `tape-arxiv23`, we performed the train/validation/test splits ourselves, where 60% of the data was allocated for training, 20% for validation, and 20% for testing. Additionally, we utilized random seeds to ensure the reproducibility of our experiments, enabling the consistent evaluation of our proposed method on the respective datasets, which can be found in our linked code repository.

### G.3 SHALLOW EMBEDDING METHODS FOR NODE FEATURE EXTRACTION

Table 9 provides an overview of the text preprocessing and feature extraction methods commonly used in graph libraries such as PyG and DGL, which are widely adopted in GNN research.

These text preprocessing and feature extraction methods facilitate the extraction of node features from the text attributes of TAG datasets, enabling the utilization of GNN models for node classification tasks. While these methods are easy to apply and computationally efficient, it is important to note that they rely on traditional language modeling techniques that may not capture the full semantic meaning in the text. This limitation can impact the expressiveness of the extracted node features and potentially affect the development of techniques for downstream tasks.

## H EXPERIMENT DETAILS

### H.1 COMPUTING ENVIRONMENT AND RESOURCES

The implementation of the proposed method utilized the PyG and DGL modules, which are licensed under the MIT License. The experiments were conducted in a computing environment with the

Table 9: Details of text preprocessing and feature extraction methods used for TAG datasets.

| Dataset | Methods | Features | Description |
|---------|---------|----------|-------------|
| Cora | BoW | 1,433 | After stemming and removing stopwords there is a vocabulary of size 1,433 unique words. All words with document frequency less than 10 were removed. |
| PubMed | TF-IDF | 500 | Each publication in the dataset is described by a TF/IDF weighted word vector from a dictionary which consists of 500 unique words. |
| ogbn-arxiv | skip-gram | 128 | The embeddings of individual words are computed by running the skip-gram model (Mikolov et al., 2013) over the MAG (Wang et al., 2020) corpus. |
| ogbn-products | BoW | 100 | Node features are generated by extracting BoW features from the product descriptions followed by a Principal Component Analysis to reduce the dimension to 100. |
| tape-arxiv23 | word2vec | 300 | The embeddings of individual words are computed by running the word2vec model. |

following specifications: LM-based experiments were performed on four NVIDIA RTX A5000 GPUs, each with 24GB VRAM. On the other hand, the GNN-based experiments were conducted on a single GPU.

## H.2 HYPERPARAMETERS

Table 10 provides an overview of the hyperparameters used for the GCN (Kipf & Welling, 2016), SAGE (Hamilton et al., 2017), and RevGAT (Li et al., 2021) models. These hyperparameters were selected based on the official OGB repository [4], and the RevGAT and language model hyperparameters follow those used in the GLEM repository [5]. It is important to note that these hyperparameters were not tuned on a per-dataset basis, but instead were used consistently across all three TAG datasets based on those from prior work, and also set consistently across both our proposed method and the baselines. This demonstrates the generality and ease of use of our method, as well as its compatibility with existing GNN baselines.

Table 10: Hyperparameters for the GCN, SAGE, and RevGAT models.

| Hyperparameters | GCN | SAGE | RevGAT |
|-----------------|-----|------|--------|
| # layers | 3 | 3 | 3 |
| hidden dim | 256 | 256 | 256 |
| learning rate | 0.01 | 0.01 | 0.002 |
| dropout | 0.5 | 0.5 | 0.75 |
| epoch | 1000 | 1000 | 1000 |
| warmup epochs | 0 | 0 | 50 |
| early stop | 50 | 50 | 50 |

## H.3 DETAILED ABLATION STUDY

We conducted a detailed ablation study on the ogbn-arxiv dataset to assess the impact of different sources of node features. The study focused on three types of node features: original text features ($h_{\mathrm{orig}}$), explanation as features ($h_{\mathrm{expl}}$), and predictions as features ($h_{\mathrm{pred}}$). We systematically removed one of these features at a time while keeping the other components unchanged in our model.

The results of the ablation study are illustrated in Figure 4. The figure presents the performance of the model when each type of node feature is removed. It is observed that using the full set of features

---

[4] https://github.com/snap-stanford/ogb
[5] https://github.com/AndyJZhao/GLEM

| Ablation | GCN | SAGE | RevGAT |
|---|---|---|---|
| Full | $0.7520 \pm 0.0003$ | $0.7672 \pm 0.0007$ | $0.7750 \pm 0.0012$ |
| - $h_{\text{orig}}$ | $0.7471 \pm 0.0007$ | $0.7433 \pm 0.0005$ | $0.7656 \pm 0.0038$ |
| - $h_{\text{expl}}$ | $0.7506 \pm 0.0011$ | $0.7528 \pm 0.0024$ | $0.7693 \pm 0.0033$ |
| - $h_{\text{pred}}$ | $0.7519 \pm 0.0019$ | $0.7605 \pm 0.0008$ | $0.7686 \pm 0.0051$ |

Figure 4: Effect of node features. We study the effects of different sources of node features on the `ogbn-arxiv` dataset, *i.e.,* original text features ($h_{\text{orig}}$), explanation as features ($h_{\text{expl}}$) and predictions as features ($h_{\text{pred}}$), by removing one of them in turn from our model while keeping the other components unchanged.

yields the best performance, while leaving out any of the features leads to a drop in performance. However, the extent of the performance drop may vary depending on the specific GNN model used.

This ablation study provides additional insights to complement the findings presented in section 5.3. While Table 3 compared the performance of using the full set of features versus using just one of them, this ablation study specifically focuses on comparing the performance of using the full set of features versus leaving one of them out. Although the experimental design differs, the overall message conveyed remains consistent, emphasizing the significance of considering all the various sources of node features for achieving optimal performance in node classification tasks.

## I    EFFECT OF LM FINETUNING

We conduct an ablation study on `ogbn-arxiv` to explore the impact of language model (LM) fine-tuning. Specifically, we aim to address the following research questions (RQs):

- RQ1: Is fine-tuning the LM necessary?
- RQ2: Is it necessary to use different LMs for encoding the original text and explanations?

To address these questions, we examine three settings: 1) Without Fine-Tuning: Utilizing a pre-trained LM to encode the original text and the explanations without any fine-tuning. 2) Fine-Tuning (Same LM): Fine-tuning a single LM for both the original text and the explanations. 3) Fine-Tuning (Different LMs): Fine-tuning two separate LMs, one for the original text and another for the explanations.

Table 11: Effect of LM finetuning on `ogbn-arxiv`

| LM | MLP | GCN | SAGE | RevGAT |
|---|---|---|---|---|
| Without Fine-Tuning | $0.5797 \pm 0.0217$ | $0.4178 \pm 0.1148$ | $0.4507 \pm 0.0529$ | $0.7507 \pm 0.0189$ |
| Fine-Tuning (Same LM) | $0.7566 \pm 0.0015$ | $0.7442 \pm 0.0012$ | $0.7676 \pm 0.0032$ | $0.7728 \pm 0.0014$ |
| Fine-Tuning (Different LMs) | $0.7587 \pm 0.0015$ | $0.7520 \pm 0.0003$ | $0.7672 \pm 0.0007$ | $0.7750 \pm 0.0012$ |

Our observations include:

For RQ1: Table 11 underscores the importance of fine-tuning the LM. It reveals a marked decline in performance without fine-tuning, compared with the settings where the LM is fine-tuned.

For RQ2: Fine-tuning, whether with the same LM or with different LMs, yields similar outcomes, with a slight advantage for using two distinct LMs. However, the marginal difference suggests that our approach could be simplified and expedited by utilizing a single LM.

## J    EFFECT OF DIFFERENT LMS

To access the influence of different LMs, we expand our investigation beyond deberta-base. Specifically, following the approach taken in SimTAG (Duan et al., 2023), we include two additional widely-used LMs from the MTEB (Muennighoff et al., 2022) leaderboard. The selection is based on

their model size and performance in classification and retrieval tasks: all-roberta-large-v1 (Reimers & Gurevych, 2019) and e5-large (Wang et al., 2022). The outcomes of our study are detailed in

Table 12: Effect of different LMs on `ogbn-arxiv`

| LM | MLP | GCN | SAGE | RevGAT |
|---|---|---|---|---|
| deberta-base | 0.7587 ± 0.0015 | 0.7520 ± 0.0003 | 0.7672 ± 0.0007 | 0.7750 ± 0.0012 |
| all-roberta-large-v1 | 0.7587 ± 0.0003 | 0.7412 ± 0.0015 | 0.7695 ± 0.0008 | 0.7737 ± 0.0004 |
| e5-large | 0.7595 ± 0.0015 | 0.7443 ± 0.0021 | 0.7688 ± 0.0010 | 0.7730 ± 0.0006 |

Table 12. Notably, our model exhibits insensitivity to the choice of a specific LM, underscoring its robustness to variations in LM selection.

## K  MEMORY UTILIZATION

Table 13 presents the memory utilization for experiments conducted on the `ogbn-arxiv` dataset.

Table 13: Memory Usage on `ogbn-arxiv` dataset with DeBERTa-base ad LM backbone and RevGAT as GNN backbone for comparision of different training paradigms of fusing LMs and GNNs, including our proposed method and the state-of-the-art GLEM method. All experiments are performed on 4 NVIDIA RTX A5000 24GB GPUs with a batch size of 36.

| Model | Memory | | Accuracy |
|---|---|---|---|
| | LM | GNN | |
| Pure LM | 8,834 MB | – | 0.7361 ± 0.0004 |
| GNN w/ shallow feature | – | 4,430 MB | 0.7083 ± 0.0017 |
| LM-based GLEM | 11,064 MB | 8,112 MB | 0.7657 ± 0.0029 |
| LLM-based TAPE (Ours) | 8,834 MB | 4,430 MB | 0.7750 ± 0.0012 |

There is a trade-off between memory consumption and accuracy. Our model appears to be the most efficient in terms of memory-to-accuracy ratio. It does not require more memory than the pure LM or GNN with shallow feature models, yet it delivers the best accuracy.

## L  GLEM

Zhao et al. (2022) evaluated GLEM on the `ogbn-arxiv` dataset. We extended our evaluation of GLEM with the `Cora` and `PubMed` datasets for a more comprehensive comparison with our method. Results are reported in Table 14

Table 14: GLEM (Zhao et al., 2022)

| Dataset | GCN | SAGE | RevGAT |
|---|---|---|---|
| Cora | 0.8732 ± 0.0066 | 0.8801 ± 0.0054 | 0.8856 ± 0.006 |
| PubMed | 0.9469 ± 0.0010 | 0.9459 ± 0.0018 | 0.9471 ± 0.002 |
| ogbn-arxiv | 0.7593 ± 0.0019 | 0.7550 ± 0.0024 | 0.7697 ± 0.0019 |

