# OpenReview forum: "Harnessing Explanations: LLM-to-LM Interpreter for Enhanced Text-Attributed Graph Representation Learning"
_ICLR.cc/2024/Conference — ICLR 2024 poster_

### Official Review · Reviewer_Dm45 · 2023-10-20

**Soundness:** 2 fair
**Presentation:** 3 good
**Contribution:** 2 fair
**Rating:** 6
**Confidence:** 3

**Summary:**

This paper addresses the challenge of leveraging Large Language Models (LLMs) such as GPT to enhance the performance of Graph Neural Networks (GNNs) on Text-Attributed Graphs (TAGs). The authors propose an innovative approach called GRAPHTEXT, which combines LLMs' textual modeling abilities with GNNs' structural learning capabilities. The key innovation lies in using explanations generated by LLMs as features to boost GNN performance on downstream tasks.

**Strengths:**

1. The proposed method presents a novel and effective way to integrate the power of LLMs, like GPT, with GNNs to handle text-attributed graphs. By using LLMs to generate explanations and converting them into informative features for GNNs, it bridges the gap between textual information and graph structure, enabling more sophisticated graph reasoning.

2. The experimental results demonstrate that the proposed method achieves state-of-the-art performance on well-established TAG datasets, including Cora, PubMed, ogbn-arxiv, and a newly introduced dataset, arXiv-2023. This underscores the effectiveness of the proposed approach in improving the accuracy of downstream tasks on TAGs.

3.The proposed method not only enhances performance but also significantly speeds up training. It achieves a 2.88 times improvement over the closest baseline on the ogbn-arxiv dataset. This computational efficiency is crucial for practical applications and scalability.

**Weaknesses:**

1. The time analysis and money estimation are lacking. The paper utilizes the chatgpt API, nonetheless, there is a large restraint on the word limitation per day, and the cost of one dataset should also be taken into consideration.

2. The robustness of the prompt is lacking. The paper proposed a single prompt for the node classification. Nonetheless, is another similar prompt can achieve a similar performance.

3. There may lack of ablation study on the LM. Current LM focuses on BERT, while other LM are ignored. Recent paper demonstrates that they can achieve satisfying performance with only SentenceBert Embedding

4. There lack some experimental results in table 1, especially Giant.

[1] Duan, Keyu, et al. "Simteg: A frustratingly simple approach improves textual graph learning." arXiv preprint arXiv:2308.02565 (2023).

**Questions:**

See the weakness.

---

> ### Author Response · Authors · 2023-11-19
> **Response to Reviewer Dm45 (Part 1/2)**
>
> We thank the reviewer for the insightful comments on our work. Please, find our answers to your comments and questions below.
>
> > **Reviewer:** The time analysis and money estimation are lacking. The paper utilizes the chatgpt API, nonetheless, there is a large restraint on the word limitation per day, and the cost of one dataset should also be taken into consideration.
>
> **Authors:** We acknowledge the importance of a comprehensive analysis of time and cost, given the constraints of the ChatGPT API.  We incorporated it in appendix D.10 (in blue) in our revised manuscript.
> Time and Cost Implications: Our primary dataset, ogbn-arxiv, with 169,343 nodes and 1,166,243 edges, serves as a representative case for our approach. On average, our input sequences consist of approximately 285 tokens, while the output sequences comprise around 164 tokens. For the ChatGPT-3.5 Turbo API, priced at $\textdollar$0.0015 per 1,000 input tokens and $\textdollar$0.002 per 1,000 output tokens, with a token per minute rate limit of 90,000, the monetary estimation for ogbn-arxiv is as follows:
>
> Cost = ((285 * 0.0015) / 1000 + (164 * 0.002) / 1000) * 169,343 ≈ 128 USD
>
> Considering the token rate limit, we estimate the deployment time as follows:
>
> Time = 169,343/ (90,000/285) ≈ 536 min ≈ 9h
>
> **Cost-Effective Alternatives**: Additionally, we have explored cost-effective alternatives, such as leveraging open-source LLMs like llama2. The use of llama2 is entirely free, and the querying process to llama2-13b-chat takes approximately 16 hours when utilizing 4 A5000 GPUs (24GB).
>
> **Efficiency through Single Query and Reuse**: Our method requires only one query to the LLM, with predictions and explanations stored for subsequent use. This not only enhances efficiency but also minimizes the number of API calls, contributing to cost-effectiveness.
>
> **Dataset Sharing**: We also make the GPT responses available for public use.
>
> > **Reviewer:** The robustness of the prompt is lacking. The paper proposed a single prompt for the node classification. Nonetheless, is another similar prompt can achieve a similar performance.
>
> **Authors:** We have extensively explored the influence of various prompts, as outlined in Table 8 and Table 9 (in blue) in the appendix. Our findings reveal a positive correlation between the zero-shot accuracy of the LLM and the overall accuracy of our method, i.e. the higher the zero-shot prediction score, the better the accuracy of TAPE. Besides, despite employing different prompt designs, our approach consistently maintains similar accuracy, ranging from 0.7660 to 0.7750, demonstrating the robustness of the prompt.
>
> > **Reviewer:** There may lack of ablation study on the LM. Current LM focuses on BERT, while other LM are ignored. Recent paper demonstrates that they can achieve satisfying performance with only SentenceBert Embedding
>
> **Authors:** We thank the referee for pointing out the necessity of a more comprehensive ablation study regarding the LM. In response, we ran additional experiments involving different LMs to assess their impact on TAPE.
>
> Before revealing our findings, we would like to address a recent paper, SimTeG [1], which uses SentenceBert and might have been referenced by the reviewer. SimTeG indeed demonstrates satisfactory results with SentenceBert Embedding. However, it is crucial to acknowledge that, similar to our model, their approach entails fine-tuning the LM-SentenceBert, with raw text and node labels. Additionally, SimTeG highlights the seamless integration of our method with theirs. A claim substantiated through experimental testing, showing that SimTeG+TAPE with Ensembling establishes a new SOTA performance on the ogbn-arxiv dataset.
>
> Moving forward, we extended our investigation beyond the deberta-base model by studying the effect of different LMs on ogbn-arxiv. In alignment with the approach taken in SimTeG [1], we incorporated two additional widely-used LMs from the MTEB leaderboard: all-roberta-large-v1 and e5-large. The detailed outcomes of our study are presented in Table 13 (in blue). Notably, our model demonstrates insensitivity to the choice of a specific LM, emphasizing its robustness to variations in LM selection.

---

> > ### Author Response · Authors · 2023-11-19
> > **Response to Reviewer Dm45 (Part 2/2)**
> >
> > > **Reviewer:** There lack some experimental results in table 1, especially Giant.
> >
> > **Authors:** We acknowledge the reviewer's observation. Our initial comparison with GIANT was conducted solely on the ogbn-arxiv dataset, as the original GIANT paper did not include experiments on Cora/PubMed, etc. In response to this feedback, we extended our comparison by applying the GIANT model to additional datasets using their official code. The comprehensive results are now presented in Table 1 (in blue). We appreciate the suggestion for a more thorough comparison across datasets. Notably, our results demonstrate consistent outperformance against GIANT on all these datasets, providing substantial support for the effectiveness of our method.
> >
> > **References**
> >
> > [1] Duan, Keyu, et al. "Simteg: A frustratingly simple approach improves textual graph learning." arXiv preprint arXiv:2308.02565 (2023).

---

> > > ### Comment · Reviewer_Dm45 · 2023-11-20
> > > **Thanks for your response**
> > >
> > > Thanks for your response. I have updated the score accordingly.

---

> > > > ### Author Response · Authors · 2023-11-21
> > > > **Response to Reviewer Dm45**
> > > >
> > > > Dear Reviewer,
> > > >
> > > >
> > > >
> > > > Thank you very much for your time reviewing our answer, and for updating your score.
> > > >
> > > >
> > > >
> > > > Best regards,
> > > >
> > > > The authors

---

### Official Review · Reviewer_HEW1 · 2023-10-23

**Soundness:** 2 fair
**Presentation:** 2 fair
**Contribution:** 2 fair
**Rating:** 5
**Confidence:** 5

**Summary:**

This paper proposes a framework for representation learning on text-attributed networks. The authors first prompt the LLM to obtain explanations and predictions for each node based on its textual information. Then they finetune language models on different textual attributes and obtain feature embeddings. Finally, the graph neural network adopts the learned embeddings from the second step to conduct the final prediction.

**Strengths:**

1. The author proposes a method to augment LM with LLM-generated features.
2. The paper is easy to follow.

**Weaknesses:**

1. Lack of baselines. Many important baselines that conduct learning with language models on TAGs including GraphFormers [1] and Patton [2] are missing in the experimental section.
2. The theorem is not specific to this problem. The theorem in 4.4 is not particular for LLM and LM for TAGs, and needs strong assumptions.
3. Lack of evaluation tasks. The paper claims to do representation learning on TAGs but only evaluates the node classification task. It would be better to add experiments on other tasks such as link prediction to evaluate the quality of the representations.
4. Lack of ablation studies. 1) If we need to finetune the LM in step 2 or not?; 2) Why do we need to have different LM for original text encoding and explanation encoding?
5. Limit novelty. While I appreciate the introduction of the new arxiv-2023 dataset, the technique contribution of this work is very limited, which basically introduces LLM to conduct data augmentation for node classification on TAGs.

[1] J. Yang, et al. GraphFormers: GNN-nested Transformers for Representation Learning on Textual Graph. NeurIPs 2021.

[2] B. Jin, et al. Patton: Language Model Pretraining on Text-Rich Networks. ACL 2023.

**Questions:**

Please refer to the comments raised in the weakness section.

---

> ### Author Response · Authors · 2023-11-19
> **Response to Reviewer HEW1 (Part 1/3)**
>
> We thank the reviewer for the examination of our work and the thoughtful comments provided. Kindly find our responses to the raised comments and questions below.
>
> > **Reviewer:** Lack of baselines. Many important baselines that conduct learning with language models on TAGs including GraphFormers [1] and Patton [2] are missing in the experimental section.
>
> **Authors:** We appreciate the reviewer’s valuable feedback and acknowledge the concern regarding the absence of certain baselines in our experimental section, particularly GraphFormers [1] and Patton [2]. We would like to clarify the reason for this absence.
>
> Our primary focus in this work is the node-prediction task in TAG. For this specific task, we concentrated on the most widely used datasets, i.e. ogbn-arxiv and ogbn-products. Additionally, recognizing the high popularity of Cora and Pubmed as node-classification datasets, we augmented these datasets by collecting text information, title and abstract, to create two additional TAGs (also made publicly available). We further introduced a new dataset, arxiv-2023, ensuring it is not part of the training set of GPT-3.5.
>
> In the context of ogbn-arxiv and ogbn-products, we chose to compare our method against GLEM [3] and GIANT [5], recognized as the strongest baselines for these datasets. While we understand the concern about the absence of GraphFormers and Patton, we believe our study offers a robust comparison by evaluating against the strongest baselines within the constraints of the available datasets.
>
> Specifically addressing GraphFormers, it was primarily designed for link prediction tasks, making a direct comparison less suitable due to differences in task objectives. Similarly, Patton involves pre-training on Microsoft Academic Graph (MAG) or Amazon e-commerce networks, followed by fine-tuning in a few-shot manner, making it also challenging for a direct comparison.
>
> Moreover, it is important to note that the datasets used in the GraphFormers and Patton papers do not overlap with the datasets in our study. Consequently, we are unable to report from their papers the performance on our datasets.
>
> Nevertheless, to address the referee’s concern, we conducted additional experiments within the rebuttal period. We extended our experiment to include MAG-Economics, a 40-way node classification task, to facilitate a comparison with Patton. The results, reported in Table 15, demonstrate that our technique TAPE outperforms GraphFormers and Patton by a significant margin of at least +13% accuracy on the MAG-Economics dataset.
>
> In the revised version, we will commit to extending the comparative analysis to other datasets such as MAG-Mathematics, MAG-Geology, Amazon-Clothes, and Amazon-Sports. We believe this comprehensive evaluation across multiple datasets will address the gap mentioned by the reviewer and provide a more holistic comparison with existing baselines.
>
> We thank the reviewer for the insightful comment, which contributes to the enhancement of the experimental section of our work.
>
>
> > **Reviewer:** The theorem is not specific to this problem. The theorem in 4.4 is not particular for LLM and LM for TAGs, and needs strong assumptions.
>
> **Authors:** While it is true that Theorem 4.4 could be applied in other contexts, we found it valuable to provide a rigorous foundation for the intuitions guiding the proposed technique.
>
> Our intention was to reinforce the understanding of why the technique is effective, beyond solely relying on experimental results. Specifically, the theorem underscores the importance of fidelity in explanations with LLM's reasoning, and emphasizes the non-redundancy between LLM and LM. These aspects constitute the key assumptions under which the technique was designed to operate successfully.

---

> > ### Author Response · Authors · 2023-11-19
> > **Response to Reviewer HEW1 (Part 2/3)**
> >
> > > **Reviewer:** Lack of evaluation tasks. The paper claims to do representation learning on TAGs but only evaluates the node classification task. It would be better to add experiments on other tasks such as link prediction to evaluate the quality of the representations.
> >
> > **Authors:** Thank you for your feedback. We have extended our experimental evaluation to include link prediction tasks as suggested.
> > Our experiments involve fine-tuning the LM for link prediction, using the resulting original text encoding and explanation encoding as node features to train a downstream GNN specifically for link prediction.
> > The results, detailed in Table 16, showcase substantial enhancements achieved by TAPE compared to GraphFormers, Patton, and GNN with shallow feature methods, i.e. without LLM-based features. This underscores the versatility and applicability of our approach beyond node classification.
> > Recognizing the need for more comprehensive evaluation tasks, we commit to include these results into the revised version.
> >
> > We once again express our gratitude to the reviewer for the valuable comment. We are confident that these enhancements will further contribute to the overall quality of our paper.
> >
> >
> >
> >
> >
> >
> >
> >
> >
> > > **Reviewer:** Lack of ablation studies. 1) If we need to finetune the LM in step 2 or not?; 2) Why do we need to have different LM for original text encoding and explanation encoding?
> >
> > **Authors:** Thank you for highlighting the need for a more comprehensive ablation study. In response to the query, we conducted additional experiments specifically on the ogbn-arxiv dataset to investigate the impact of LM fine-tuning.
> > Addressing the reviewer’s specific points:
> > 1) The significance of fine-tuning the LM in step 2: As illustrated in Table 12, the row labeled "without fine-tuning" starkly exhibits inferior performance when using a pre-trained LM without fine-tuning, compared to the last two rows showcasing the positive impact of LM fine-tuning.
> > 2) The necessity of having different LMs for the original text and explanations: Table 12, with rows "fine-tuning (same LM)" versus "fine-tuning (different LMs)," demonstrates that training the same or distinct LMs for encoding the original text and the explanations yields similar results. There is a slight advantage for using two LMs, but the marginal difference suggests that our approach can be simplified and expedited by employing one LM.
> > We appreciate the reviewer's insight, which helps simplify the presented technique.

---

> > ### Author Response · Authors · 2023-11-19
> > **Response to Reviewer HEW1 (Part 3/3)**
> >
> > > **Reviewer:** Limit novelty. While I appreciate the introduction of the new arxiv-2023 dataset, the technique contribution of this work is very limited, which basically introduces LLM to conduct data augmentation for node classification on TAGs.
> >
> > **Authors:** We reproduce below our general response regarding the importance and novelty of our work:
> >
> > With the emerging dominance of LLMs, it has become critical to integrate LLMs with GNNs for text- and graph-based tasks such as TAG, knowledge graphs, and text reasoning. Our work stands out as a pioneering effort in this direction.
> >
> > The task of representation learning on TAGs involves integrating the raw text with graphs in a robust and scalable way. Existing literature can be broadly categorized into two paradigms: 1) End-to-end or joint training of LM and GNN, exemplified by approaches like GLEM [3] and DRAGON [4]; and 2) Two-stage training, where LM is initially trained followed by GNN training. Our approach belongs to the second paradigm, inheriting its efficiency and ease of plugging into existing GNN pipelines, but incorporates an LLM-to-LM interpreter to extract and interpret the LLM's useful prior knowledge.
> >
> > The first approach is generally considered more attractive because the LM is trained to directly leverage graph structure. However, it often involves complex language and graph architectures, encounters training instabilities, and demands significant GPU resources, particularly challenging with LLMs. In this context, an important contribution of our work is the finding that with the help of our LLM-to-LM interpreter, the second paradigm of two-stage training can achieve equal or better performance than end-to-end or joint training, with much greater efficiency. In particular, our LLM-to-LM interpreter allows the valuable prior knowledge from LLMs to be embedded into GNN features, and our work shows that this does not sacrifice performance but greatly improves efficiency, compared to joint training.
> >
> > Specifically, we decompose the text+graph combination into two distinct parts: a LLM+LM interpreter harnessing LLM capabilities but challenging to fine-tune, and a GNN using text features fine-tuned for the graph task. This decomposition significantly stabilizes the training process, as each part is independently executed — prompting LLM, fine-tuning LM with node labels, and supervised training of the GNN.
> >
> > This approach shares similarities with the training of LLMs like ChatGPT or Llama2, which involve multiple independent training steps, including self-supervised learning, fine-tuning, and reinforcement learning. This departure from optimizing in a one-shot end-to-end manner, known for its instability, has proven highly successful in NLP, and now in TAG demonstrated with our technique through extensive experiments.
> >
> > Lastly, we believe that LLMs have disrupted the field of GNNs,but the current focus on end-to-end or joint LM+GNN training creates near-prohibitive scalability issues when applied to LLMs, making it harder to reach the potential of LLMs for graph learning. Our work takes a significant step by showing that LLM-to-LM interpreters can be used to embed useful knowledge from LLMs into GNN pipelines in a simple and efficient way, without sacrificing performance. As a matter of fact, since the introduction of our work, we have witnessed a surge in similar techniques published on ArXiv, many of which have achieved top rankings on the OGB leaderboard. This underscores the significance of our paper as the pioneer work in this line of research, which is poised to grow in the coming years.
> >
> >
> > **References**
> >
> > [1] J. Yang, et al. GraphFormers: GNN-nested Transformers for Representation Learning on Textual Graph. NeurIPS 2021.
> >
> > [2] B. Jin, et al. Patton: Language Model Pretraining on Text-Rich Networks. ACL 2023.
> >
> > [3] Jianan Zhao, Meng Qu, Chaozhuo Li, Hao Yan, Qian Liu, Rui Li, Xing Xie, and Jian Tang. Learning on large-scale text-attributed graphs via variational inference. ICLR, 2023
> >
> > [4] Michihiro Yasunaga, Antoine Bosselut, Hongyu Ren, Xikun Zhang, Christopher D Manning, Percy S Liang, and Jure Leskovec. Deep bidirectional language-knowledge graph pretraining. NeurIPS, 2022.
> >
> > [5] Chien, Eli, et al. Node feature extraction by self-supervised multi-scale neighborhood prediction.

---

> ### Comment · Reviewer_HEW1 · 2023-11-20
>
> I would like to thank the authors for their detailed response and their added experiments. I'm still not fully convinced by the novelty of this work, but I'm happy to raise my score.

---

> > ### Author Response · Authors · 2023-11-21
> > **Response to Reviewer HEW1**
> >
> > Dear Reviewer,
> >
> >
> >
> > We appreciate very much your acknowledgment of our response, considering the new experiments, and for raising your score.
> >
> >
> >
> > If novelty is defined by addressing the existing limitations and challenges of current SOTA techniques, we are confident that this work has indeed fulfilled this definition in light of a new line of arXiv publications built upon our work :)
> >
> >
> >
> > Best regards,
> >
> > The authors

---

### Official Review · Reviewer_NoEV · 2023-11-01

**Soundness:** 3 good
**Presentation:** 3 good
**Contribution:** 2 fair
**Rating:** 6
**Confidence:** 5

**Summary:**

This paper proposes a method to solve TAG problems. The method uses an LLM to generate category prediction and explanation of the target node, then the explanation is used as an enhancing part of the node feature. An LM is used to encode the raw text feature and the additional explanation into hidden space. A GNN predictor receives the hidden features, the prediction of LLM, and the shallow embeddings to give the final classification results. Experiments on several TAGs show the effectiveness of the method.

**Strengths:**

1. The paper is clearly written and easy to follow.
2. Bagging LLM prediction, LM features and shallow features is reasonable for the node classification task.
3. The experiments show the method can achieve SOTA performance on several benchmark datasets.

**Weaknesses:**

1. The main concern is that the method has little to do with graph. It seems like an application attempt of LLMs on natural language tasks, and TAG is just a scenario. Thus the contribution of the method is limited.
2. Since the Debera need fully fine-tuning, training the method cost much more memory than pure GNN methods.

**Questions:**

See weaknesses.

---

> ### Author Response · Authors · 2023-11-19
> **Response to Reviewer NoEV**
>
> We thank the reviewer very much for the careful reading and comments regarding our work. Please, see below our answer to the raised comments/questions.
>
> > **Reviewer:** The main concern is that the method has little to do with graph. It seems like an application attempt of LLMs on natural language tasks, and TAG is just a scenario. Thus the contribution of the method is limited.
>
> **Authors:** While it is true that the first part of our technique has no direct connection to graph topology, and focuses on node features, the overall methodology is strongly motivated towards addressing the challenges inherent in the TAG problem, an active research area propelled by the advent of language models.
>
> As advocated in our global response, our approach is deeply rooted in addressing the challenges associated with the joint training of GNNs and LMs. The task of representation learning on TAGs involves integrating the raw text with graphs in a robust and scalable way. Existing literature can be broadly categorized into two paradigms: 1) End-to-end or joint training of LM and GNN, exemplified by approaches like GLEM [1] and DRAGON [2]; and 2) Two-stage training, where LM is initially trained followed by GNN training. Our approach belongs to the second paradigm, inheriting its efficiency and ease of plugging into existing GNN pipelines, but incorporates an LLM-to-LM interpreter to extract and interpret the LLM's useful prior knowledge.
>
> The first approach is generally considered more attractive because the LM is trained to directly leverage graph structure. However, it often involves complex language and graph architectures, encounters training instabilities, and demands significant GPU resources, particularly challenging with LLMs. In this context, an important contribution of our work is the finding that with the help of our LLM-to-LM interpreter, the second paradigm of two-stage training can achieve equal or better performance than end-to-end or joint training, with much greater efficiency. In particular, our LLM-to-LM interpreter allows the valuable prior knowledge from LLMs to be embedded into GNN features, and our work shows that this does not sacrifice performance but greatly improves efficiency, compared to joint training.
>
> Therefore, our work should be viewed as a strategic response aimed at overcoming these challenges. It goes beyond being a mere application of LLMs to TAG; rather, it represents a novel, efficient, and robust approach to addressing tasks that involve both text and graphs.
>
> We acknowledge that we may not have sufficiently conveyed the important message that our work aims to resolve the intricate issues tied to LM+GNN training for TAG. We are committed to amending the paper to provide greater clarity on this aspect for the reader.
>
>
> > **Reviewer:** Since the Deberta need fully fine-tuning, training the method cost much more memory than pure GNN methods.
>
> **Authors:** We appreciate the reviewer's observation regarding the increased memory usage in our approach compared to pure GNN methods, as illustrated in Figure 2. It is important to note that learning on TAG inherently demands the integration of LMs and GNNs. Established methods like GIANT [5], GLEM [1], GraphFormers [3], and Patton [4], among others, all involve LM in their processes. Therefore, a meaningful comparison of memory cost should be drawn against these baselines rather than pure GNNs.
>
> Recognizing the inherent tradeoff between cost and accuracy, we highlight that despite the increased resource requirements, our method TAPE significantly boosts the accuracy of the pure GNN method from 70.83% to 77.50%. We firmly believe that the substantial performance improvement justifies the accompanying rise in memory costs. Notably, these elevated resource demands are manageable, even with relatively cost-effective academic GPU resources like our A5000 GPU (24GB).
> We greatly appreciate the reviewer's observation regarding memory usage, which prompted us to conduct a more in-depth analysis, as outlined in Table 14 (in blue), showing that TAPE provides both best performance, and comparable or better running time than other LM-based methods.
>
>
>
>
> **Reference**
>
> [1] Jianan Zhao, Meng Qu, Chaozhuo Li, Hao Yan, Qian Liu, Rui Li, Xing Xie, and Jian Tang. Learning on large-scale text-attributed graphs via variational inference. ICLR, 2023
>
> [2] Michihiro Yasunaga, Antoine Bosselut, Hongyu Ren, Xikun Zhang, Christopher D Manning, Percy S Liang, and Jure Leskovec. Deep bidirectional language-knowledge graph pretraining. NeurIPS, 2022.
>
> [3] J. Yang, et al. GraphFormers: GNN-nested Transformers for Representation Learning on Textual Graph. NeurIPS 2021.
>
> [4] B. Jin, et al. Patton: Language Model Pretraining on Text-Rich Networks. ACL 2023.
>
> [5] Chien, Eli, et al. Node feature extraction by self-supervised multi-scale neighborhood prediction.

---

> > ### Author Response · Authors · 2023-11-21
> > **Appreciation for Your Review and Awaited Feedback**
> >
> > Dear Reviewer,
> >
> > Thank you once again for your comprehensive review of our work.
> >
> > As the deadline for discussion draws near, we are eager to receive your feedback on our response to your questions.
> >
> > Additionally, we welcome any further questions or comments you may have.
> >
> > Best regards,
> >
> > The authors

---

### Author Response · Authors · 2023-11-19
**Summary of the revised manuscript**

We thank the Reviewers and the Area Chair for dedicating their valuable time and effort to review our paper.

We summarize below the main contribution of the paper as well as the new experiments.

**Contribution**

With the emerging dominance of LLMs, it has become critical to integrate LLMs with GNNs for text- and graph-based tasks such as TAG, knowledge graphs, and text reasoning. Our work stands out as a pioneering effort in this direction.

The task of representation learning on TAGs involves integrating the raw text with graphs in a robust and scalable way. Existing literature can be broadly categorized into two paradigms: 1) End-to-end or joint training of LM and GNN, exemplified by approaches like GLEM [1] and DRAGON [2]; and 2) Two-stage training, where LM is initially trained followed by GNN training. Our approach belongs to the second paradigm, inheriting its efficiency and ease of plugging into existing GNN pipelines, but incorporates an LLM-to-LM interpreter to extract and interpret the LLM's useful prior knowledge.

The first approach is generally considered more attractive because the LM is trained to directly leverage graph structure. However, it often involves complex language and graph architectures, encounters training instabilities, and demands significant GPU resources, particularly challenging with LLMs. In this context, an important contribution of our work is the finding that with the help of our LLM-to-LM interpreter, the second paradigm of two-stage training can achieve equal or better performance than end-to-end or joint training, with much greater efficiency. In particular, our LLM-to-LM interpreter allows the valuable prior knowledge from LLMs to be embedded into GNN features, and our work shows that this does not sacrifice performance but greatly improves efficiency, compared to joint training.

Specifically, we decompose the text+graph combination into two distinct parts: an LLM+LM interpreter harnessing LLM capabilities but challenging to fine-tune, and a GNN using text features fine-tuned for the graph task. This decomposition significantly stabilizes the training process, as each part is independently executed — prompting LLM, fine-tuning LM with node labels, and supervised training of the GNN.

This approach shares similarities with the training of LLMs like ChatGPT or Llama2, which involve multiple independent training steps, including self-supervised learning, fine-tuning, and reinforcement learning. This departure from optimizing in a one-shot end-to-end manner, known for its instability, has proven highly successful in NLP, and now in TAG demonstrated with our technique through extensive experiments.

Lastly, we believe that LLMs have disrupted the field of GNNs, but the current focus on end-to-end or joint LM+GNN training creates near-prohibitive scalability issues when applied to LLMs, making it harder to reach the potential of LLMs for graph learning. Our work takes a significant step by showing that LLM-to-LM interpreters can be used to embed useful knowledge from LLMs into GNN pipelines in a simple and efficient way, without sacrificing performance. As a matter of fact, since the introduction of our work, we have witnessed a surge in similar techniques published on ArXiv, many of which have achieved top rankings on the OGB leaderboard. This underscores the significance of our paper as the pioneer work in this line of research, which is poised to grow in the coming years.

We are committed to amending the paper to provide greater clarity on this aspect for the reader.

**New Experiments**

In response to reviewers' valuable feedback during the rebuttal phase, we have conducted additional experiments to address all reviewers' concerns:
- Expanded evaluation tasks, showcasing TAPE’s generalization to link prediction (Table 12)
- Inclusion of more baseline methods such as GraphFormers and Patton. (Table 15 & 16)
- Ablation studies on the effect of fine-tuning LM (Table 12)
- Ablation study on the choice of LM (Table 13)
- Evaluation of the robustness of the prompt (Table 9)
- Comprehensive comparison with GIANT, addressing the lack of some experimental results (Table 1)

Please, find the new experiments highlighted in blue color in the pdf file.

We hope that the revision of the manuscript with the new experiments and our point-to-point responses have clarified the reviewers’ concerns. We are happy to answer any additional questions to clarify further.

Thank you again for your dedicated effort and time to review the paper.

Best regards,

The authors



**References**

[1] Jianan Zhao, Meng Qu, Chaozhuo Li, Hao Yan, Qian Liu, Rui Li, Xing Xie, and Jian Tang. Learning on large-scale text-attributed graphs via variational inference. ICLR, 2023

[2] Michihiro Yasunaga, Antoine Bosselut, Hongyu Ren, Xikun Zhang, Christopher D Manning, Percy S Liang, and Jure Leskovec. Deep bidirectional language-knowledge graph pretraining. NeurIPS, 2022.

---

### Meta-Review · Area_Chair_wibt · 2023-12-15

**Metareview:**

This paper explores using LLMs as feature extractors to enhance the performance of GNNs on text attributed graphs. The authors have used extensive empirical studies to demonstrate the effectiveness of the proposed method, establishing new state-of-the-art results. While some reviewers raised concerns regarding the technical novelty of the proposed method, I believe there is still significant contribution as being the first of its kind.

**Justification For Why Not Higher Score:**

There is indeed limited technical contribution, as reviewers pointed out.

**Justification For Why Not Lower Score:**

The proposed method is the first of its kind and is empirically effective.

---

### Decision · Program_Chairs · 2024-01-16

Accept (poster)